# Learning Debiased Classifier with Biased Committee

Nayeong Kim    Sehyun Hwang    Sungsoo Ahn    Jaesik Park    Suha Kwak

Pohang University of Science and Technology (POSTECH), South Korea
{nayeong.kim, sehyun03, sungsoo.ahn, jaesik.park, suha.kwak}@postech.ac.kr

## Abstract

Neural networks are prone to be biased towards spurious correlations between classes and latent attributes exhibited in a major portion of training data, which ruins their generalization capability. We propose a new method for training debiased classifiers with no spurious attribute label. The key idea is to employ a committee of classifiers as an auxiliary module that identifies bias-conflicting data, *i.e.*, data without spurious correlation, and assigns large weights to them when training the main classifier. The committee is learned as a bootstrapped ensemble so that a majority of its classifiers are biased as well as being diverse, and intentionally fail to predict classes of bias-conflicting data accordingly. The consensus within the committee on prediction difficulty thus provides a reliable cue for identifying and weighting bias-conflicting data. Moreover, the committee is also trained with knowledge transferred from the main classifier so that it gradually becomes debiased along with the main classifier and emphasizes more difficult data as training progresses. On five real-world datasets, our method outperforms prior arts using no spurious attribute label like ours and even surpasses those relying on bias labels occasionally. Our code is available at `https://github.com/nayeong-v-kim/LWBC`.

## 1  Introduction

Most supervised learning algorithms for classification rely on the empirical risk minimization (ERM) principle [40]. However, ERM has been known to cause a learned classifier to be biased toward spurious correlations between predefined classes and latent attributes that appear in a majority of training data [12]. In the case of hair color classification, for example, when most people with `blond-hair` (*i.e.*, target class) are `female` (*i.e.*, latent attribute) in a dataset, a classifier learned by ERM exploits `female` as a shortcut for the classification due to its spurious correlation with `blond-hair`, and often mis-classifies non-blonde-haired women as `blond-hair` in consequence. We call data with such spurious correlations and holding a majority of training data *bias-guiding samples*, and the other *bias-conflicting samples*, respectively. The issue of model bias has often been addressed by exploiting explicit spurious attribute labels [22, 28, 36, 2, 39, 38, 45] or knowledge about bias types given a priori [3]. However, these methods are impractical because such supervision and prior knowledge are costly, and the methods demand extensive post hoc analysis.

Hence, a body of research has been conducted for learning debiased classifiers with no additional label for spurious attributes [42, 27, 29, 32, 23, 26]. A common approach in this line of work is to employ an intentionally biased classifier as an auxiliary module [29, 32, 23, 26]. In this approach, samples that the biased classifier has trouble handling are regarded as bias-conflicting ones and assigned large weights when used for training the main classifier to reduce the effect of bias-guiding counterparts. Although it has driven remarkable success, this approach has drawbacks due to the use of a single biased classifier. First, the quality of the biased classifier could vary by hyper-parameters [29] and its initial parameter values [11]. Further, data that the biased classifier fails to handle could include not only bias-conflicting samples but also bias-guiding ones, which differs by the quality of the classifier.

36th Conference on Neural Information Processing Systems (NeurIPS 2022).

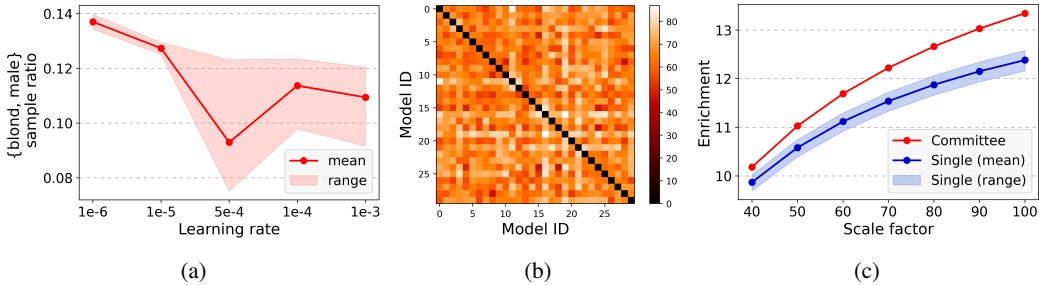

(a)          (b)          (c)

Figure 1: Analysis on the instability of a single biased classifier in mining and weighting bias-conflicting samples. The experiments are conducted on the CelebA dataset, in which samples with `blond` and `male` attributes are bias-conflicting. (a) The ratio of bias-conflicting samples to all incorrectly predicted by a single biased classifier. The ratio highly fluctuates by the learning rate of the classifier and varies up to 4%p due to different initialization even with a fixed learning rate, meaning that a single biased classifier is sensitive to hyper-parameters. (b) Disagreement on predictions of biased classifiers. For pairs of biased classifiers initialized differently, we measure the number of bias-conflicting samples for which the classifiers predict differently. The results suggest that individual biased classifiers are sensitive to initialization. (c) Comparisons between a single biased classifier and a committee of biased classifiers in terms of enrichment [29]. Higher enrichment implies more precise mining and weighting of bias-conflicting samples; the formal definition of enrichment is given in Appendix **??**. The committee clearly outperforms the single biased classifier in terms of the enrichment.

These drawbacks limit the reliability and performance of debiasing methods depending on a single biased classifier, as demonstrated in Figure 1.

To overcome these limitations, we propose a new method using a committee of biased classifiers as the auxiliary module, coined *learning with biased committee* (LWBC). LWBC identifies bias-conflicting samples and determines their weights through consensus on their prediction difficulty within the committee. To this end, the committee is built as a bootstrapped ensemble, *i.e.*, each of its classifiers is trained from a randomly sampled subset of the entire training dataset. This strategy not only guarantees the diversity among the classifiers, but also lets a majority of the classifiers be biased since random subsets of training data are highly likely to be dominated by bias-guiding samples. Accordingly, a majority of the committee tends to classify bias-guiding samples correctly and fail to deal with bias-conflicting ones. The consensus on prediction difficulty within the committee thus gives a strong cue for identifying and weighting bias-conflicting samples. Also, using the consensus of multiple classifiers enables LWBC to be robust to the varying quality of individual classifiers and consequently to focus more precisely on bias-conflicting samples, as shown in Figure 1.

Moreover, unlike the biased classifier trained independently of the main classifier in the previous work, the committee in LWBC is trained with knowledge of the main classifier as well as the random subsets of training data to serve the main classifier better. Specifically, the knowledge is distilled in the form of classification logits of the main classifier [18], and each classifier of the committee utilizes the knowledge as pseudo labels of training data other than its own training set. We expect that this strategy allows the committee to become debiased gradually so that it does not give large weights to easy bias-conflicting samples, *i.e.*, those already well handled by the main classifier, and focuses more on difficult ones. Note that, even with this strategy, the classifiers of the committee are still biased differently due to their different training sets with ground-truth labels.

Finally, we further improve the proposed method by adopting a self-supervised representation as the frozen backbone of the committee and the main classifier. Since self-supervised learning is not dependent on class labels, it is less affected by the spurious correlations between classes and latent attributes, leading to a robust and less-biased representation. Also, by installing the committee and the main classifier on top of the representation, the classifiers can be implemented efficiently in both space and time while enjoying the rich and bias-free features given by the backbone.

LWBC is validated extensively on five real-world datasets. It substantially outperforms existing methods using no bias label and even occasionally surpasses previous arts demanding bias labels. We also demonstrate that all of the main components, *i.e.*, the use of the committee, its training with

knowledge transfer, and the self-supervised learning, contribute to the outstanding performance. The main contribution of this paper is four-fold:

- We present LWBC, a new approach to learning a debiased classifier with no spurious attribute label. The use of consensus within the committee allows LWBC to address limitations of previous work relying on a single biased classifier.

- We propose to learn the committee using knowledge of the main classifier, unlike the previous work whose auxiliary modules do not consider the main classifier.

- We investigate the potential of self-supervised learning for debiasing, and find that it is a solid yet unexplored baseline for the task.

- LWBC demonstrates superior performance on five real-world datasets. It outperforms existing methods using no additional supervision like ours and even surpasses those relying on spurious attribute labels occasionally.

## 2 Related work

### 2.1 Debiasing with prior knowledge on bias

A majority of previous work on learning debiased classifiers mitigates the bias by leveraging explicit labels on bias types [22, 28, 36, 2, 39, 38, 45]. For instance, Kim et al. [22] build a debiased model to classify digits by leveraging RGB color labels explicitly in the Colored MNIST dataset. Other methods are designed to handle predefined domain-specific bias types (*e.g.*, texture bias in ImageNet) [41, 5, 3].

Recent methods try to reduce annotation costs for bias supervision by only utilizing a small set of bias-labeled data. For instance, Nam et al. [33] and Jung et al. [21] train auxiliary bias predictor with a small set of bias-labeled data and assign pseudo bias labels to the entire training set using its prediction. These methods have two inherent limitations. First, they cannot handle biases whose types are not predefined in training. Second, manually annotating all bias types is often expensive and laborious. Even obtaining a small set of bias labels could be expensive since bias labels existing in a real-world dataset is often long-tailed [30] or multi-labeled [16].

### 2.2 Debiasing without spurious attribute labels

Recent debiasing methods without any bias supervision [19, 8, 32, 26, 23, 29, 7, 1, 37, 25, 34] are based on the assumption that attributes of malignant bias are easier to learn than those of target classes. Through this assumption, previous methods identify the amount of biases within the samples and train a robust models by emphasizing bias-conflicting samples. To identify biases of samples, previous work leverage a high gradient of the latent vector [19, 8] or employ an intentionally biased classifier as an auxiliary module [32, 23, 26] by using generalized cross-entropy (GCE) loss [44]. Liu et al. [29] consider misclassified samples as bias-conflicting samples, and Creager et al. [7] derive bias-guiding and bias-conflicting partition that maximally violates the invariance principle. On the other hand, Bahng et al. [3] propose a method tailored to the texture bias, building a debiased model to learn independent features from a biased model in which receptive field size is limited to capture the texture bias in image classification.

While LWBC is also based on the aforementioned assumption, we firstly propose to identify bias-conflicting samples and up-weight them through the consensus of a committee of the biased classifiers. By exploiting the consensus of multiple classifiers, LWBC is robust against the varying quality of individual classifiers and allows to learn debiased classifiers more reliably and effectively.

## 3 Proposed method

We propose a new method that learns a debiased classifier with a committee of biased classifiers, dubbed LWBC. It first learns a feature representation with self supervision, which is used as the frozen backbone providing rich and bias-free features to downstream modules (Section 3.1). Next, it trains a committee of $m$ auxiliary classifiers $f_1, f_2, \ldots, f_m$ and the main classifier $g$ on top of

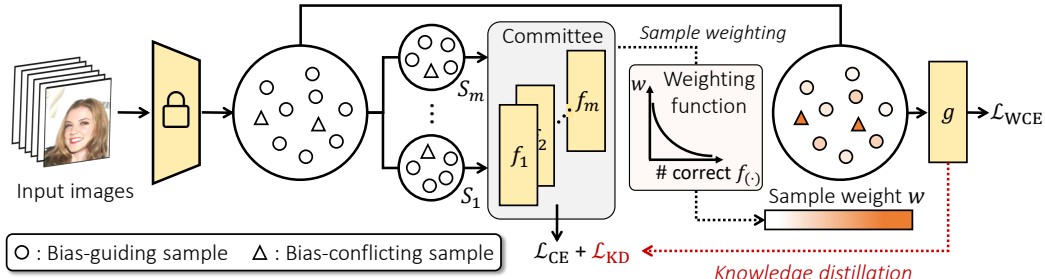

Figure 2: The overall pipeline of LWBC. It adopts a frozen backbone trained by self-supervised learning. A committee of auxiliary classifiers $f_i$ is trained on top of the backbone; a random subset of training data is assigned to each classifier of the committee as labeled data for supervised learning (Eq. (1)). The committee determines the weight of each training sample based on consensus of its members, *i.e.*, the number of members that correctly predict the class of the sample (Eq. (2)). The main classifier $g$ is trained using the weights by minimizing the weighted cross entropy loss (Eq. (3)). In turn, knowledge of the main classifier is transferred to the committee through knowledge distillation (Eq. (4)).

the self-supervised representation (Section 3.2); thanks to the self-supervised representation, the classifiers are designed concisely, using only two fully-connected layers for each.

In a nutshell, the committee identifies bias-conflicting samples and assigns them large weights to reduce the effect of bias-guiding samples during the training of the main classifier. To this end, the committee is trained as a bootstrapped ensemble of classifiers so that a majority of its classifiers are biased as well as diverse, and intentionally fail to predict classes of bias-conflicting samples accordingly. Hence, the consensus within the committee on prediction difficulty of a sample (*e.g.*, the number of classifiers that fail to predict its class label) indicates how much likely the sample is bias-conflicting, and is used to compute weights for training samples. Moreover, the committee is trained also with knowledge of the main classifier so that it gradually becomes debiased along with the main classifier and emphasizes more difficult samples as training progresses. Note that the committee is an auxiliary module used only in training and thus does not impose additional computation or memory footprint in testing.

The overall process of LWBC is illustrated conceptually in Figure 2 and given formally in Algorithm 1. The following sections elaborate on each step of LWBC.

## 3.1 Self-supervised representation learning

As the feature extractor, we train a backbone network by self-supervised learning with BYOL [14] on the target dataset. During the self-supervised learning, a random patch of input image is cropped, resized to 224×224 pixels, flipped horizontally at random, and distorted by a random sequence of brightness, contrast, saturation, hue adjustments, and grayscale conversion.

A self-supervised model can capture diverse patterns shared by data without being biased towards a particular class even the training set is biased. We empirically demonstrate that adopting a self-supervised representation leads to a model less biased compared with a supervised representation.

Although the representation offers rich and less-biased features, the main classifier can be still biased when it is trained by ERM. In other words, the self-supervised representation alone is not enough to learn a debiased model, and the need to explore a debiasing method for a classifier arises. Hence, we propose LWBC, a new debiasing method illustrated in the next section.

## 3.2 Learning a debiased classifier with a biased committee

First, we randomly sample $m$ subsets of the same size, denoted by $\mathcal{S}_1, ..., \mathcal{S}_m$, from the entire training dataset $\mathcal{D}$ with replacement. Then $m$ auxiliary classifiers $f_1, ..., f_m$ of the committee are initialized randomly, and each of the subsets $\mathcal{S}_l$ is assigned to each auxiliary classifier $f_l$ as its training data.

---

**Algorithm 1** Learning a debiased classifier with a biased committee

---

**input** training set $\mathcal{D}$, batch size $b$, size of the committee $m$, learning rate $\eta$, scale hyper-parameter $\alpha$, balancing hyper-parameter $\lambda$, total number of iterations $t$, number of warm-up iterations $t_w$

1: Draw $m$ random subsets $\mathcal{S}_1, ..., \mathcal{S}_m$ of $\mathcal{D}$.
2: Initialize auxiliary classifiers of the committee $\{f_l(x; \theta_l)\}_{l=1}^m$.
3: Initialize the main classifier $g(x; \phi)$.
4: **for** $j = 1, ..., t_w$ **do**
5:     Draw a mini-batch $\mathcal{B} = \{(x_i, y_i)\}_{i=1}^b$ from $\mathcal{D}$.
6:     $\theta_l \leftarrow \theta_l - \eta \nabla_{\theta_l} \mathcal{L}_{\text{CE}} \quad \forall l = 1, ..., m$            Eq. (1)
7: **end for**
8: **for** $j = t_w + 1, ..., t$ **do**
9:     Draw a mini-batch $\mathcal{B} = \{(x_i, y_i))\}_{i=1}^b$ from $\mathcal{D}$.
10:    $w(x_i) = 1/(\sum_{l=1}^m \mathbb{1}(f_l(x_i) = y_i)/m + \alpha) \quad \forall x_i \in \mathcal{B}$     Eq. (2)
11:    $\phi \leftarrow \phi - \eta \nabla_\phi \mathcal{L}_{\text{WCE}}$                                      Eq. (3)
12:    $\theta_l \leftarrow \theta_l - (1 - \lambda)\eta \nabla_{\theta_l} \mathcal{L}_{\text{CE}} - \lambda \eta \nabla_{\theta_l} \mathcal{L}_{\text{KD}} \quad \forall l = 1, ..., m$    Eq. (1), Eq. (4)
13: **end for**

---

The first step of LWBC is warm-up training of the committee; this is required to ensure that the committee is capable of identifying and weighting bias-conflicting samples at the beginning of the main training process. Given a mini-batch $\mathcal{B}$ at each warm-up iteration, the committee is trained by minimizing the cross-entropy loss below:

$$\mathcal{L}_{\text{CE}} = \sum_{l=1}^m \sum_{(x,y) \in \mathcal{S}_l \cap \mathcal{B}} \text{CE}(f_l(x), y). \tag{1}$$

Since each subset is sampled from the training set dominated by bias-guiding samples, a majority of auxiliary classifiers are also biased. At the same time, the classifiers are diverse due to their difference in initialization and training data.

After the warm-up stage, the main classifier and the committee are trained while interacting with each other. First, the main classifier is trained by the weighted cross entropy loss with the entire training set, where the sample weights are computed by considering consensus within the committee on prediction difficulty of the samples. Since a majority of auxiliary classifiers have trouble to handle bias-conflicting samples, we identify and weight bias-conflicting samples based on the number of auxiliary classifiers whose predictions are correct for the samples. The weight function $w$ is given by

$$w(x) = \frac{1}{\sum_{l=1}^m \mathbb{1}(f_l(x) = y)/m + \alpha}, \tag{2}$$

where $m$ is the size of the committee, $f_l$ means the $l$-th classifier of the committee, and $\alpha$ is a scale hyper-parameter. The weight reflects how much the sample $x$ is likely to be bias-conflicting and decreases rapidly when the number of correctly predicting classifiers increases. Then we train the main classifier $g$ with emphasis on the bias-conflicting samples through the weight function $w$. The weighted cross entropy loss $\mathcal{L}_{\text{WCE}}$ is given by

$$\mathcal{L}_{\text{WCE}} = \sum_{(x,y) \in \mathcal{B}} w(x) \cdot \text{CE}(g(x), y), \tag{3}$$

where $\mathcal{B}$ is the mini-batch.

During training, as the main classifier is gradually debiased, samples useful for debiasing the main classifier change accordingly. To focus more on bias-conflicting samples difficult for the main classifier, we inform the quality of the main classifier to the committee by distilling the knowledge of the main classifier in the form of its classification logits [18] and transferring the knowledge by minimizing the following KD loss:

$$\mathcal{L}_{\text{KD}} = \sum_{l=1}^m \sum_{(x,y) \in \mathcal{B} \setminus \mathcal{S}_l} \text{KL}\left(\text{softmax}\left(\frac{g(x)}{\tau}\right), \text{softmax}\left(\frac{f_l(x)}{\tau}\right)\right), \tag{4}$$

where $\tau$ is a temperature parameter. Note that we apply $\mathcal{L}_{\text{KD}}$ to the complement set of $\mathcal{S}_l$ to avoid auxiliary classifiers in the committee being identical to each other. By interacting with the main

Table 1: GUIDING, UNBIASED, and CONFLICTING metrics (%) for the CelebA dataset. For methods without spurious attribute labels, we mark the best and the second-best performance in **bold** and underline, respectively.

| Method | Spurious attribute label | CelebA HairColor | | | CelebA HeavyMakeup | | |
|---|---|---|---|---|---|---|---|
| | | GUIDING | UNBIASED | CONFLICTING | GUIDING | UNBIASED | CONFLICTING |
| Group DRO [36] | ✓ | 87.46 | $85.43_{\pm0.53}$ | $83.40_{\pm0.67}$ | 79.52 | $64.88_{\pm0.42}$ | $50.24_{\pm0.68}$ |
| EnD [38] | ✓ | 94.97 | $91.21_{\pm0.22}$ | $87.45_{\pm1.06}$ | 98.16 | $75.93_{\pm1.31}$ | $53.70_{\pm5.24}$ |
| CSAD [45] | ✓ | 91.19 | 89.36 | 87.53 | 82.32 | 67.88 | 53.44 |
| ERM | ✗ | 87.98 | $70.25_{\pm0.35}$ | $52.52_{\pm0.19}$ | $\underline{90.25}$ | $62.00_{\pm0.02}$ | $33.75_{\pm0.28}$ |
| LfF [32] | ✗ | 87.24 | $\underline{84.24}_{\pm0.37}$ | $\underline{81.24}_{\pm1.38}$ | 86.92 | $66.20_{\pm1.21}$ | $\underline{45.48}_{\pm4.43}$ |
| SSL+ERM | ✗ | $\mathbf{94.15}_{\pm0.57}$ | $80.48_{\pm0.91}$ | $66.79_{\pm2.20}$ | $\mathbf{93.00}_{\pm0.49}$ | $\underline{66.30}_{\pm1.15}$ | $39.50_{\pm2.47}$ |
| LWBC | ✗ | $\underline{90.57}_{\pm2.15}$ | $\mathbf{88.90}_{\pm1.55}$ | $\mathbf{87.22}_{\pm1.14}$ | $86.42_{\pm0.82}$ | $\mathbf{70.29}_{\pm1.14}$ | $\mathbf{51.28}_{\pm5.74}$ |

Table 2: UNBIASED and WORST-GROUP metrics (%) for the CelebA dataset. We also report the difference between UNBIASED and WORST-GROUP as GAP. For methods without spurious attribute labels, we mark the best and the second-best performance in **bold** and underline, respectively.

| Method | Backbone network | Spurious attribute label | CelebA HairColor | | |
|---|---|---|---|---|---|
| | | | UNBIASED | WORST-GROUP | GAP |
| Group DRO [36] | Resnet50 | ✓ | $93.1_{\pm0.21}$ | $88.5_{\pm1.16}$ | 4.6 |
| SSA [33] | Resnet50 | ✓ | $92.8_{\pm0.11}$ | $89.8_{\pm1.28}$ | 3.0 |
| ERM | Resnet50 | ✗ | $\mathbf{95.6}$ | 47.2 | 48.4 |
| CVaR DRO [27] | Resnet50 | ✗ | 82.4 | 64.4 | 18.0 |
| LfF [32] | Resnet50 | ✗ | 86.0 | 70.6 | 15.4 |
| EIIL [7] | Resnet50 | ✗ | $\underline{91.9}$ | $\underline{83.3}$ | 8.6 |
| JTT [29] | Resnet50 | ✗ | 88.0 | 81.1 | $\underline{6.9}$ |
| SSL+ERM | Resnet18 | ✗ | $80.5_{\pm0.9}$ | $38.5_{\pm4.1}$ | 42.0 |
| LWBC | Resnet18 | ✗ | $88.9_{\pm1.6}$ | $\mathbf{85.5}_{\pm1.4}$ | $\mathbf{3.4}$ |

classifier, the committee gradually becomes debiased along with the main classifier. Hence, samples correctly predicted by the main classifier are less weighted and those with incorrect predictions are more weighted by the committee.

After the warm-up training, the main classifier and the auxiliary classifiers are alternately updated with a given mini-batch at each iteration. First, we forward every sample in a mini-batch to each auxiliary classifier and then compute weights of the samples using predictions of the auxiliary classifiers (Eq. (2)). With the weights, the main classifier is updated by Eq. (3) and the knowledge of the updated main classifier is transferred to the auxiliary classifiers by Eq. (4). Thus, the auxiliary classifiers are updated by minimizing both losses of Eq. (1) and Eq. (4):

$$\mathcal{L}_{\text{committee}} = (1 - \lambda)\mathcal{L}_{\text{CE}} + \lambda\mathcal{L}_{\text{KD}}, \qquad (5)$$

where $\lambda$ is a balancing hyper-parameter.

## 4 Experiments

### 4.1 Setup

**Implementation details.** We adopt ResNet-18 [15] as the self-supervised model and train it following the strategy of BYOL [14] on the target dataset. Since the color information is key feature for `HairColor` classification on the CelebA dataset, we vary only brightness and contrast for data augmentation of color distortion when training the model on CelebA. Since NICO and BAR are very small datasets, we initialize the self-supervised model with ImageNet [9] pretrained parameters; except for the experiments using BAR and NICO, the self-supervised model is trained from scratch. We use the self-supervised ResNet-18 as a backbone network except for the last fully connected layer. We set the batch size to {64, 64, 128, 256}, learning rate to {1e-3, 1e-3, 1e-4, 6e-3}, the size of the committee $m$ to {30, 30, 30, 40}, the size of subset $\mathcal{S}_l$ to {10, 10, 80, 300}, $\lambda$ to {0.9, 0.6, 0.6, 0.6},

Table 3: VALIDATION, UNBIASED, and TEST metrics (%) evaluated on the ImageNet-9 and ImageNet-A datasets. For methods without spurious attribute labels, we mark the best and the second-best performance in **bold** and underline, respectively.

| Method | Spurious attribute label | ImageNet-9 | | ImageNet-A |
| --- | --- | --- | --- | --- |
| | | VALIDATION | UNBIASED | TEST |
| StylisedIN [13] | ✓ | $88.4_{\pm 0.5}$ | $86.6_{\pm 0.6}$ | $24.6_{\pm 1.4}$ |
| LearnedMixin [6] | ✓ | $64.1_{\pm 4.0}$ | $62.7_{\pm 3.1}$ | $15.0_{\pm 1.6}$ |
| RUBi [5] | ✓ | $90.5_{\pm 0.3}$ | $88.6_{\pm 0.4}$ | $27.7_{\pm 2.1}$ |
| ERM | ✗ | $90.8_{\pm 0.6}$ | $88.8_{\pm 0.6}$ | $24.9_{\pm 1.1}$ |
| Biased (BagNet18) [4] | ✗ | $67.7_{\pm 0.3}$ | $65.9_{\pm 0.3}$ | $18.8_{\pm 1.15}$ |
| ReBias [3] | ✗ | $91.9_{\pm 1.7}$ | $90.5_{\pm 1.7}$ | $29.6_{\pm 1.6}$ |
| LfF [32] | ✗ | 86.0 | 85.0 | 24.6 |
| CaaM [42] | ✗ | **95.7** | **95.2** | 32.8 |
| SSL+ERM | ✗ | $94.18_{\pm 0.07}$ | $93.18_{\pm 0.04}$ | $34.21_{\pm 0.49}$ |
| LWBC | ✗ | $94.03_{\pm 0.23}$ | $93.04_{\pm 0.32}$ | **$35.97_{\pm 0.49}$** |

and $\tau$ to {1, 1, 1, 2.5}, respectively for {BAR, NICO, Imagenet-9, CelebA}, and $\alpha$ to 0.02. Note that we run LWBC on 3 random seeds and report the average and the standard deviation.

**Evaluation metrics.** Six metrics are adopted for evaluation. VALIDATION / TEST: average accuracy on validation / test splits. GUIDING: average accuracy on bias guiding samples per class. CONFLICTING: average accuracy on bias conflicting samples per class. UNBIASED: average of GUIDING and CONFLICTING per class. WORST-GROUP: minimum average accuracy of group; we can group the validation and test samples by the target and the bias.

### 4.2 Datasets

**CelebA.** CelebA [31] is a dataset for face recognition where each sample is labeled with 40 attributes. Following the experiment configuration suggested by Nam et al. [32], we focus on `HairColor` and `HeavyMakeup` attributes that are spuriously correlated with `Gender` attributes, *i.e.*, most of the CelebA images with `blond-hair` are women. As a result, the biased model suffers from performance degradation when predicting `HairColor` attribute on males. Therefore, we use `HairColor` as the target attribute and `Gender` as a spurious attribute, the same as `HeavyMakeup`.

**ImageNet-9.** ImageNet-9 [20] is a subset of ImageNet [35] containing nine super-classes. Following the setting adopted by Bahng et al. [3], we conduct experiments with 54,600 training images and 2,100 validation images. ImageNet-9 has been known to have a correlations between object class and image texture. We follow the evaluation scheme adopted by Bahng et al. [3], and we report the unbiased accuracy of the validation set, which is computed as average accuracy on every object-texture combination.

**ImageNet-A.** ImageNet-A [17] contains real-world images misclassified by an ImageNet-trained ResNet 50 [15]. Since such failures are caused when the model too heavily relies on colors, textures, and backgrounds, ImageNet-A could be regarded as a bias-conflicting set w.r.t. various ImageNet biases. This dataset is used only for evaluating a model trained on ImageNet-9.

**BAR.** The Biased Action Recognition (BAR) dataset [32] is a real-world image dataset intentionally designed to exhibit spurious correlations between human action and place on its images. Originally the training set of BAR consists of only bias-guiding samples, and its test set consists of only bias-conflicting samples. In our setting, we use 10% of the original BAR training set as validation and set the bias-conflicting ratio of the training set to 1%.

**NICO.** NICO [16] is a real-world dataset for simulating out-of-distribution image classification scenarios. Following the setting used by Wang et al. [42], we use an animal subset of NICO, which is labeled with 10 object and 10 context classes for evaluating the debiasing methods. The training set consists of 7 context classes per object class and they are long-tailed distributed (e.g., dog images are more frequently coupled with the 'on grass' context than any of the other 6 contexts). The validation and test sets consist of 7 seen context classes and 3 unseen context classes per object class. We verify the ability of debiasing a model from object-context correlations through evaluation on NICO.

Table 4: CONFLICTING, VALIDATION, and TEST metrics (%) evaluated on the BAR dataset (a) and the NICO dataset (b). For methods without spurious attribute labels, we mark the best and the second-best performance in **bold** and underline, respectively.

(b) NICO dataset

| Method | Spurious attribute label | NICO | |
| --- | --- | --- | --- |
| | | VALIDATION | TEST |
| Cutout [10] | ✓ | 43.69 | 43.77 |
| RUBi [5] | ✓ | 43.86 | 44.37 |
| IRM [2] | ✓ | 40.62 | 41.46 |
| Unshuffle [39] | ✓ | 43.15 | 43.00 |
| REx [24] | ✓ | 41.00 | 41.15 |
| ERM | ✗ | 43.77 | 42.61 |
| CBAM [43] | ✗ | 42.15 | 42.46 |
| ReBias [3] | ✗ | 44.92 | 45.23 |
| LfF [32] | ✗ | 41.83 | 40.18 |
| CaaM [42] | ✗ | 46.38 | 46.62 |
| SSL+ERM | ✗ | $55.63_{\pm 0.54}$ | $52.24_{\pm 0.27}$ |
| LWBC | ✗ | $\mathbf{56.05}_{\pm 0.45}$ | $\mathbf{52.84}_{\pm 0.31}$ |

(a) BAR dataset

| Method | Spurious attribute label | BAR |
| --- | --- | --- |
| | | CONFLICTING |
| ERM | ✗ | $35.32_{\pm 0.46}$ |
| ReBias [3] | ✗ | $37.02_{\pm 0.26}$ |
| LfF [32] | ✗ | $48.15_{\pm 0.93}$ |
| LDD [26] | ✗ | $52.31_{\pm 1.00}$ |
| SSL+ERM | ✗ | $60.88_{\pm 0.80}$ |
| LWBC | ✗ | $\mathbf{62.03}_{\pm 0.74}$ |

Table 5: Ablation studies using WORST-GROUP metric (%) on the CelebA HairColor dataset. We study the impact of learning from a single biased classifier (row 2), learning by committee (row 3), and transferring the knowledge of the main classifier (row 4). We mark the best performance in **bold**.

| Method | | | | | CelebA HairColor |
| --- | --- | --- | --- | --- | --- |
| SSL | ERM | Single | Committee | KD | WORST-GROUP |
| ✓ | ✓ | ✗ | ✗ | ✗ | $38.5_{\pm 4.1}$ |
| ✓ | ✗ | ✓ | ✗ | ✗ | $64.1_{\pm 2.4}$ |
| ✓ | ✗ | ✗ | ✓ | ✗ | $81.3_{\pm 2.3}$ |
| ✓ | ✗ | ✗ | ✓ | ✓ | $\mathbf{85.5}_{\pm 1.4}$ |

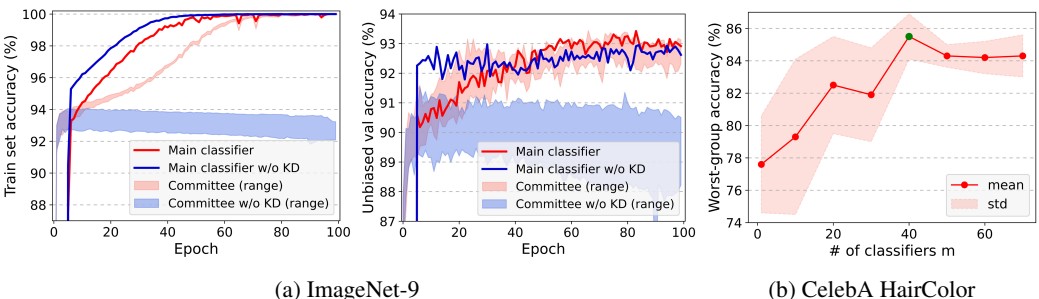

(a) ImageNet-9

(b) CelebA HairColor

Figure 3: Effect of knowledge distillation (Eq. (4)) and the size of the committee $m$. (a) Accuracy of the main classifier and the committee with or without KD loss in ImageNet-9 train set (*left*) and unbiased validation set (*right*). (b) WORST-GROUP accuracy of LWBC versus the number of classifiers within the committee, where the green dot indicates the value used in the main paper.

### 4.3 Quantitative results

LWBC shows superior classification accuracy among the methods that do not use the spurious attribute label on the five real-world datasets. In Tables 1 & 2, we observe LWBC outperforms existing debiasing methods using no spurious attribute label and shows comparable CONFLICTING performance with methods that exploit spurious attribute labels on CelebA, which reflects gender bias in the real world. Especially, the gap between the average accuracy of groups and worst-group accuracy of LWBC is much smaller than the other methods, suggesting that our model fairly predicts a sample that belongs to each group. Table 3 shows the results on ImageNet-9, which is dominated by

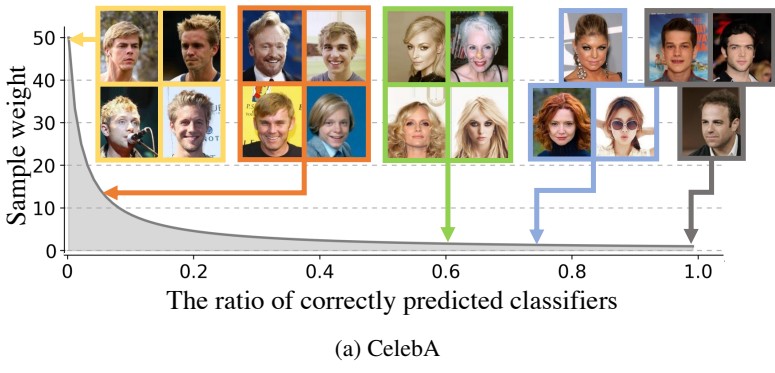

(a) CelebA

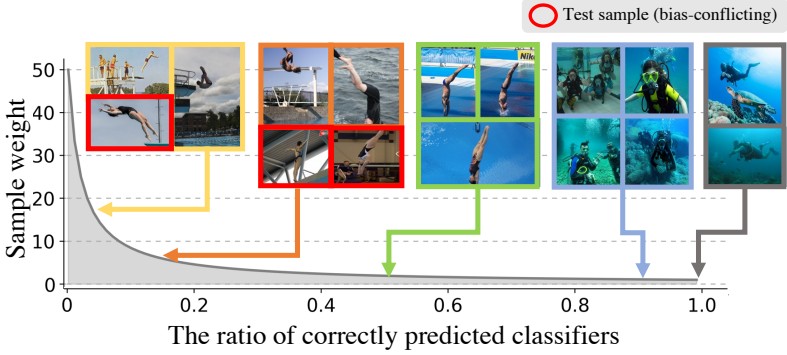

(b) BAR

Figure 4: Qualitative examples of sample weighting by LWBC on CelebA and BAR. The graph represents the weight function in Eq. (2) along with example images corresponding to specific weights. (a) The examples on the CelebA dataset. `HairColor` is the target and `Gender` is the bias. A majority of `blond-hair` images on the training set is `female`. (b) The examples of `diving` class on the BAR dataset. `Action` is the target and `Place` is the bias. A majority of `diving` images on the training set include a body of water or the surface of a body of water. The red bordered images are test samples which are bias-conflicting samples.

texture bias, and ImageNet-A, which is regarded as a bias-conflicting set of ImageNet. LWBC is 9.7% better than CaaM [43] on the ImageNet-A dataset, *i.e.*, LWBC is robust to texture bias and various ImageNet biases. Table 4a shows results on the BAR dataset. LWBC is 18.6% better than LDD, the previous state of the art. Compared with LfF [32] and LDD [26], which are debiasing methods with a single biased classifier, learning a debiased classifier with the biased committee is more effective. Table 4b shows the results on the NICO dataset. LWBC is 13.3% better than CaaM, which suggests that LWBC is better generalized to unseen spurious attributes. Note that the validation and test set of NICO have unseen context classes and are unbiased.

## 4.4 Ablation study

**Self-supervised representation as a solid baseline.** We empirically investigate the potential of self-supervised representation as a solid baseline for the debiasing task. We train a classifier on the top of the self-supervised representation by ERM. The results are denoted by 'SSL+ERM' and compared with 'ERM', which is a fully supervised classification model in Table 1, 2, 3, 4a, 4b. 'SSL+ERM' outperforms not only ERM but also the previous state-of-the-art on all the datasets except for CelebA. 'SSL+ERM' is less biased than the model trained by fully supervised learning.

**Importance of each module in LWBC.** Table 5 demonstrates through ablation studies [29, 32, 23, 26]: (1) learning from a single biased classifier, (2) learning with committee, and (3) transferring the knowledge of the main classifier. First, we train a classifier by ERM (row 1) then assign a weight value 50 to the wrongly predicted samples and 1 to other samples. Then re-training the classifier with the weights (row 2). Comparing these two results demonstrates that up-weighting scheme with a biased

classifier is effective to debiasing a classifier. Then we increase the number of biased classifiers and compute the sample weights using our weight function Eq. (2) (row 3). Learning with the committee shows a remarkable improvement in the worst-group accuracy. Moreover, knowledge distillation that enables the committee to interact with the main classifier further improves performance (row 4).

**Effectiveness of transferring knowledge of the main classifier.** Figure 3a shows the range of unbiased validation accuracy of classifiers in the committee and unbiased validation accuracy of the main classifier during training. The mean accuracy of classifiers in the committee gradually increases following the accuracy of the main classifier. Also, the accuracy of the main classifier gradually increases as training progresses. On the other hand, the accuracy of classifiers in the committee trained without KD loss does not increase or even decrease.

**Number of classifiers.** We compare the results of the main classifier trained with the different number of auxiliary classifiers. Figure 3b shows the worst-group accuracy versus the number of auxiliary classifiers $m$ on CelebA. With a single auxiliary classifier, the main classifier shows the lowest worst-group accuracy, but the accuracy increases as $m$ increases. When larger than 40, the number of classifiers has little effect on learning the main classifier.

## 4.5 Qualitative results

Figure 4 shows the graph of the weight function in Eq. (2) along with example images corresponding to specific weights. LWBC not only up-weights the bias-conflicting samples and down-weights the bias-guiding samples but also imposes fine-grained weights according to the difficulty of samples. For example, a majority of training samples of `diving` class on BAR are images of scuba diving or those of falling into the water with a blue background. In contrast, test samples of `diving` class are sport images of falling into water from a platform or springboard. As illustrated in Figure 4b, LWBC imposes fine-grained weights based on the proportion of the background in the image as well as down-weighting scuba diving images and up-weighting falling from platform images. The results demonstrate that LWBC more precisely distinguishes samples according to bias attributes.

# 5 Limitations

Self-supervised learning increases overall training complexity as it usually takes a longer time for convergence than supervised learning. Also, since the committee is designed as a bootstrapped ensemble, it involves randomness in training and thus causes noticeable performance fluctuation on CelebA, although it demonstrates stable performance on the other datasets.

# 6 Conclusion

We have proposed a new method for learning a debiased classifier with a committee of auxiliary classifiers. The committee is learned in a way that consensus on predictions of its classifiers offers a strong and reliable cue to identify and weight bias-conflicting data. The main debiased classifier is then trained with an emphasis on the bias-conflicting data to reduce the effect of bias-guiding counterparts. The committee is also trained to be debiased gradually along with the main classifier so that it highlights more challenging data as training progresses. Moreover, we demonstrated that self-supervised learning is a solid yet unexplored baseline for debiasing. Coupled with a self-supervised feature extractor, our method achieved state-of-the-art by large margins on most existing real-world datasets.

**Acknowledgments and Disclosure of Funding**

This work was supported by the NRF grants and the IITP grants funded by the Ministry of Education and the Ministry of Science and ICT, Korea (NRF-2021R1A2C3012728, 40%; NRF-2022R1A6A1A03052954, 20%; IITP-2022-0-00290, 30%; IITP-2019-0-01906, 10%).

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
