# Learning Debiased Classifier with Biased Committee

**Nayeong Kim    Sehyun Hwang    Sungsoo Ahn    Jaesik Park    Suha Kwak**

Pohang University of Science and Technology (POSTECH), South Korea
{nayeong.kim, sehyun03, sungsoo.ahn, jaesik.park, suha.kwak}@postech.ac.kr

## A  Appendix

### A.1  Implementation details

Throughout all the experiments in the paper, we adopt Adam optimizer [1] and employ no data augmentation scheme when training the main classifier and the committee. We tune all hyperparameters as well as early stop based on highest CONFLICTING for CelebA and BAR, VALIDATION for NICO and ImageNet-9 on validation set. For a fair comparison, we follow the existing validation set construction procedure, which varies for different datasets. To be specific, the validation sets are either uniformly sampled from the whole dataset (ImageNet-9 and BAR), include bias-guiding ones (NICO), or consist only of bias-conflicting ones (CelebA). The hyperparameters, their search spaces, and the performance metric used for the tuning are summarized in Table 1. The knowledge transfer from the main classifier to the committee is conducted after the warm-up training of the committee and an additional epoch for training the main classifier, which ensures that the main classifier is trained sufficiently, using the entire training set once, before transferring its knowledge. We warm-up the committee during 3 epochs on small datasets (NICO, BAR) and 5 epochs on large datasets (CelebA, ImageNet). We did not carefully tune the number of warm-up iterations and empirically found that 3-5 epochs were enough to achieve stable performance. The results of LfF [3] on NICO and ImageNet-9 are obtained by using the official code `https://github.com/alinlab/LfF`. We use a single GPU (RTX 3090 for {CelebA, BAR, NICO} and TITAN RTX for ImageNet-9) for training.

Table 1: The search spaces of hyperparameters. We mark the adopted value in **bold**.

|  | CelebA | ImageNet-9 | BAR | NICO |
|---|---|---|---|---|
| metric | CONFLICTING | VALIDATION | CONFLICTING | VALIDATION |
| set | validation | validation | validation | validation |
| Batch size | 256 | 128 | 64 | 64 |
| Learning rate | {1e-2, 4e-3, 5e-3, **6e-3**} | {1e-3, **1e-4**} | {**1e-3**, 1e-4} | {**1e-3**, 1e-4} |
| $m$ | {10, 20, 30, **40**, 50, 60, 70} | {20, **30**, 40} | {20, **30**, 40} | {20, **30**, 40} |
| $\alpha$ | {0.015, **0.02**, 0.025} | {**0.02**, 0.2} | {**0.02**, 0.2} | {**0.02**, 0.2} |
| $|\mathcal{S}_l|$ | {200, **300**, 400, 600, 800, 900, 1600} | {**80**} | {**10**, 20, 30} | {**10**, 20, 30} |
| $\lambda$ | {0.3, 0.4, 0.5, **0.6**, 0.7, 0.8} | {0.5, **0.6**, 0.7, 0.8, 0.9} | {0.5, 0.6, 0.7, 0.8, **0.9**} | {0.5, **0.6**, 0.7, 0.8, 0.9} |
| $\tau$ | {1, 1.5, 2, **2.5**, 3} | {**1**} | {**1**} | {**1**} |

36th Conference on Neural Information Processing Systems (NeurIPS 2022).

## A.2   Details of Figure 1

The bias-conflicting set of CelebA consists of both `blond-hair` & `male` samples and `non-blond-hair` & `female` samples, and we consider only `blond-hair` & `male` samples as bias-conflicting samples in Figure 1. Since the correlation between `non-blond-hair` and `male` attributes is less dominant than the correlation between `blond` and `female` attributes, a model is less biased toward `non-blond-hair` and `male` attributes. 'Single' up-weights the bias-conflicting samples to a scale factor. 'Committee' up-weights the bias-conflicting samples using the weight function in Eq. (2) and $\alpha$ equals to $1/$(scale factor). The Enrichment [30] measures how much the model more focuses on the bias-conflicting samples compared to ERM training. Specifically, it is calculated by

$$\text{Enrichment} = \frac{(\text{sum of weights of bias-conflicting samples}/\text{sum of weights})}{(\text{\# of bias-conflicting samples}/\text{\# of samples})}. \qquad (6)$$

## A.3   Ablation study

Table 7: Ablation studies using WORST-GROUP metric (%) on the CelebA HairColor dataset. We study the impact of training randomly sampled subset, learning by committee, and transferring the knowledge of the main classifier. We mark the best performance in **bold**. The search space of hyperparameters of JTT [30] on SSL features in as follows: {1, 2, 10, 20, 30, 50} for the identification epoch and {20, 50, 100} for the upweight value. Optimal hyperparamters of epoch and upweight we found are 10 and 20, respectively.

| | Method | | | | CelebA HairColor |
|---|---|---|---|---|---|
| SSL | Weight function | Training data for each committee member | Committee size (m) | KD | WORST-GROUP |
| ✓ | ERM | - | 0 | ✗ | $38.5_{\pm4.1}$ |
| ✓ | JTT [30] | Entire training dataset | 1 | ✗ | $57.7_{\pm11.5}$ |
| ✓ | Ours | Entire training dataset | 1 | ✗ | $64.1_{\pm2.4}$ |
| ✓ | Ours | Random subset | 1 | ✗ | $72.7_{\pm11.7}$ |
| ✓ | Ours | Random subset | 1 | ✓ | $77.9_{\pm3.1}$ |
| ✓ | Ours | Entire training dataset | 40 | ✗ | $78.0_{\pm4.3}$ |
| ✓ | Ours | Random subset (bootstrapped) | 40 | ✗ | $81.3_{\pm2.3}$ |
| ✓ | Ours | Random subset (bootstrapped) | 40 | ✓ | $\mathbf{85.5}_{\pm1.4}$ |

Table 8: Ablation studies using UNBIASED, CONFLICTING, and WORST-GROUP metric (%) on the CelebA HairColor dataset. We study the impact of bootstrapping from deep ensemble (row 1), ensemble with SSL (row 2), LWBC without SSL (row 3), and LWBC with SSL (row 4). We mark the best performance in **bold**.

| Method | Committee member | Training data for each committee member | Committee size (m) | SSL feature | UNBIASED | CONFLICTING | WORST-GROUP |
|---|---|---|---|---|---|---|---|
| Deep ensemble | Whole model | Entire trainset | 5 | ✗ | 83.0 | 79.2 | 77.2 |
| Ensemble w/ SSL | Two FC layers | Entire trainset | 40 | ✓ | $87.0_{\pm1.5}$ | $83.6_{\pm1.1}$ | $78.0_{\pm4.3}$ |
| LWBC w/o SSL | Two FC layers | Random subset | 40 | ✗ | $87.1_{\pm0.6}$ | $83.9_{\pm1.6}$ | $80.8_{\pm1.6}$ |
| LWBC w/ SSL | Two FC layers | Random subset | 40 | ✓ | $88.9_{\pm1.6}$ | $87.2_{\pm1.1}$ | $\mathbf{85.5}_{\pm1.4}$ |

We extend Table 5 of the main paper. The extended table can be found in Table 7, Table 8, and Table 9.

**Performance with a single biased model on SSL features.** As demonstrated in Table 7, every variant of LWBC using SSL features outperforms JTT using SSL features on CelebA. The performance of the LWBC variant using a single biased model trained on the whole training set (Table 7 row 3) significantly outperforms that of JTT on SSL features (Table 7 row 2). The main difference between the two models, which leads to the performance gap, is two fold. First, they use different weight

functions. Second, the single biased classifier of the LWBC variant produces sample weights at every iteration while the weights are computed once and fixed during training in the JTT variant.

**Impact of knowledge distillation.** We conduct two additional ablation studies with two variants of LWBC to figure out the effectiveness of KD. The first variant incorporates a single biased classifier learned on a subset of the training set (Table 7 row 4), and the second variant additionally adopts KD (Table 7 row 5). The results in Table 7 suggest that KD is useful for debiasing regardless of the use of the committee.

**Impact of bootstrapping.** In Table 8, we study the impact of bootstrapping in our method. We compare our method with deep ensemble, ensemble using SSL feature, and ours without SSL features. Note that each auxiliary classifier in the 'Deep ensemble' and 'Ensemble w/ SSL' setting learns from the same data, but they are differently initialized. Unlike LWBC, the KD loss in Eq. (2) is calculated using the entire training set for the 'Ensemble w/ SSL' experiment. Deep ensemble showed the worst performance. Using SSL features and increasing the size of the ensemble improves performance slightly, but still largely inferior to LWBC. As shown in Table 8, bootstrapping (*i.e.*, LWBC) substantially outperforms the naive ensemble regardless of the use of SSL features. In addition, using SSL features further improves performance. We believe that this is because the auxiliary classifiers trained on a subset of the training set are more diverse and biased than those trained on the entire training set.

Table 9: Ablation studies using UNBIASED, CONFLICTING, and WORST-GROUP metric (%) on the CelebA HairColor dataset. We study the impact of frozen backbone trained by supervised learning on celebA (row 1-3), supervised learning on ImageNet (row 4-6), and self-supervised learning on celebA (row 7-9). Learning by ERM (row 1, 4, 7), learning by committee (row 2, 5, 8), and transferring the knowledge of the main classifier (row 3, 6, 9). We mark the best performance in **bold**.

| Backbone | | | Method | | | CelebA HairColor | | |
|---|---|---|---|---|---|---|---|---|
| Supervised on celebA | ImageNet pretrained | Self-sup on celebA | ERM | Committee | KD | UNBIASED | CONFLICTING | WORST-GROUP |
| ✓ | ✗ | ✗ | ✓ | ✗ | ✗ | $94.6_{\pm 0.01}$ | $70.3_{\pm 0.2}$ | $45.2_{\pm 0.6}$ |
| ✓ | ✗ | ✗ | ✗ | ✓ | ✗ | $78.9_{\pm 10.0}$ | $75.2_{\pm 6.3}$ | $54.0_{\pm 27.0}$ |
| ✓ | ✗ | ✗ | ✗ | ✓ | ✓ | $80.0_{\pm 9.4}$ | $78.9_{\pm 3.1}$ | $61.1_{\pm 24.4}$ |
| ✗ | ✓ | ✗ | ✓ | ✗ | ✗ | $75.3_{\pm 2.3}$ | $60.9_{\pm 1.8}$ | $28.0_{\pm 5.9}$ |
| ✗ | ✓ | ✗ | ✗ | ✓ | ✗ | $84.2_{\pm 1.0}$ | $80.9_{\pm 0.8}$ | $68.9_{\pm 2.9}$ |
| ✗ | ✓ | ✗ | ✗ | ✓ | ✓ | $85.1_{\pm 0.6}$ | $82.4_{\pm 1.4}$ | $76.6_{\pm 4.6}$ |
| ✗ | ✗ | ✓ | ✓ | ✗ | ✗ | $80.5_{\pm 0.9}$ | $66.8_{\pm 2.2}$ | $38.5_{\pm 4.1}$ |
| ✗ | ✗ | ✓ | ✗ | ✓ | ✗ | $88.6_{\pm 1.3}$ | $84.0_{\pm 1.7}$ | $81.3_{\pm 1.4}$ |
| ✗ | ✗ | ✓ | ✗ | ✓ | ✓ | $\mathbf{88.9}_{\pm 1.6}$ | $\mathbf{87.2}_{\pm 1.1}$ | $\mathbf{85.5}_{\pm 1.4}$ |

**Impact of backbone.** In Table 9, we study the impact of a frozen backbone trained by self-supervised learning (row 7-9) compared to supervised learning (row 1-3 for ERM backbone and row 4-6 for ImageNet pretrained backbone). Within the results using the same backbone, learning with the committee and transferring the knowledge of the main classifier to the committee improve performance in all metrics compared with the ERM classifier, regardless of the backbone. Regarding the performance of the ERM classifier on top of each backbone (row 1, 4, 7), the ERM backbone leads to the best performance among the three backbones since the ERM backbone is trained with class labels. However, the ERM backbone was not useful when coupled with our method (learning with the committee and KD) dedicated to debiasing. This shows the limitation of conventional representation based on supervised learning. Comparing ImageNet pretrained backbone and self-supervised trained backbone (both are target-label-free schemes), the backbone trained by self-supervised learning is always better than the ImageNet pretrained backbone in our experiments. We believe that this is because a frozen backbone trained by self-supervised learning on a target dataset gives rich and bias-free features. Surprisingly, the main classifier learned by the committee and KD on top of ImageNet pretrained frozen backbone using the same hyper-parameters outperforms LfF [33], which demonstrates that the advantage of our method is not limited to a specific backbone network.

### A.4 Qualitative results

#### A.4.1 Class activation map

Figure 5 shows the class activation map (CAM) of the main classifier, those of auxiliary classifiers of the committee, and a consensus graph on a bias-guiding sample of CelebA, and Figure 6 shows them on a bias-conflicting sample of CelebA. We mark a classifier that correctly predicts the class of the sample in 'correct', otherwise 'incorrect'.

In Figure 5, a majority of auxiliary classifiers correctly predict the class of the bias-guiding sample, but they focus on facial appearance. Since auxiliary classifiers have a consensus on 'correct', the main classifier less focus on the sample during training.

In Figure 6, a majority of auxiliary classifiers wrongly predict the class of the bias-conflicting sample because they focus on facial appearance. Since auxiliary classifiers have a consensus on 'incorrect', the main classifier focuses more on the sample during training. The main classifier does not focus on facial appearance to correctly predict both the bias-guiding and bias-conflicting samples.

As we expected, a majority of auxiliary classifiers focus on facial appearance, *i.e.*, auxiliary classifiers exploit `gender` feature rather than `HairColor` feature to classify an image. However, the main classifier focuses more on `HairColor` feature than the auxiliary classifiers.

#### A.4.2 Qualitative examples on the NICO dataset

Figure 7 shows the graph of weight function in Eq. 2 along with example images corresponding to specific weights on the NICO dataset. As illustrated on Figure 7, LWBC not only up-weights the bias-conflicting samples and down-weights the bias-guiding samples but also imposes fine-grained weights according to difficulty of a sample.

### A.5 Comparison between single classifier and the committee in terms of the ratio of identified bias-conflicting samples

Figure 8 shows the ratio graph of bias-conflicting samples among the examples identified by a committee according to the number of correct classifiers in the committee, and that of a single biased classifier as a blue point. The ratio by the committee is much higher than that of a single biased classifier. This result shows that the biased committee precisely identified bias-conflicting samples.

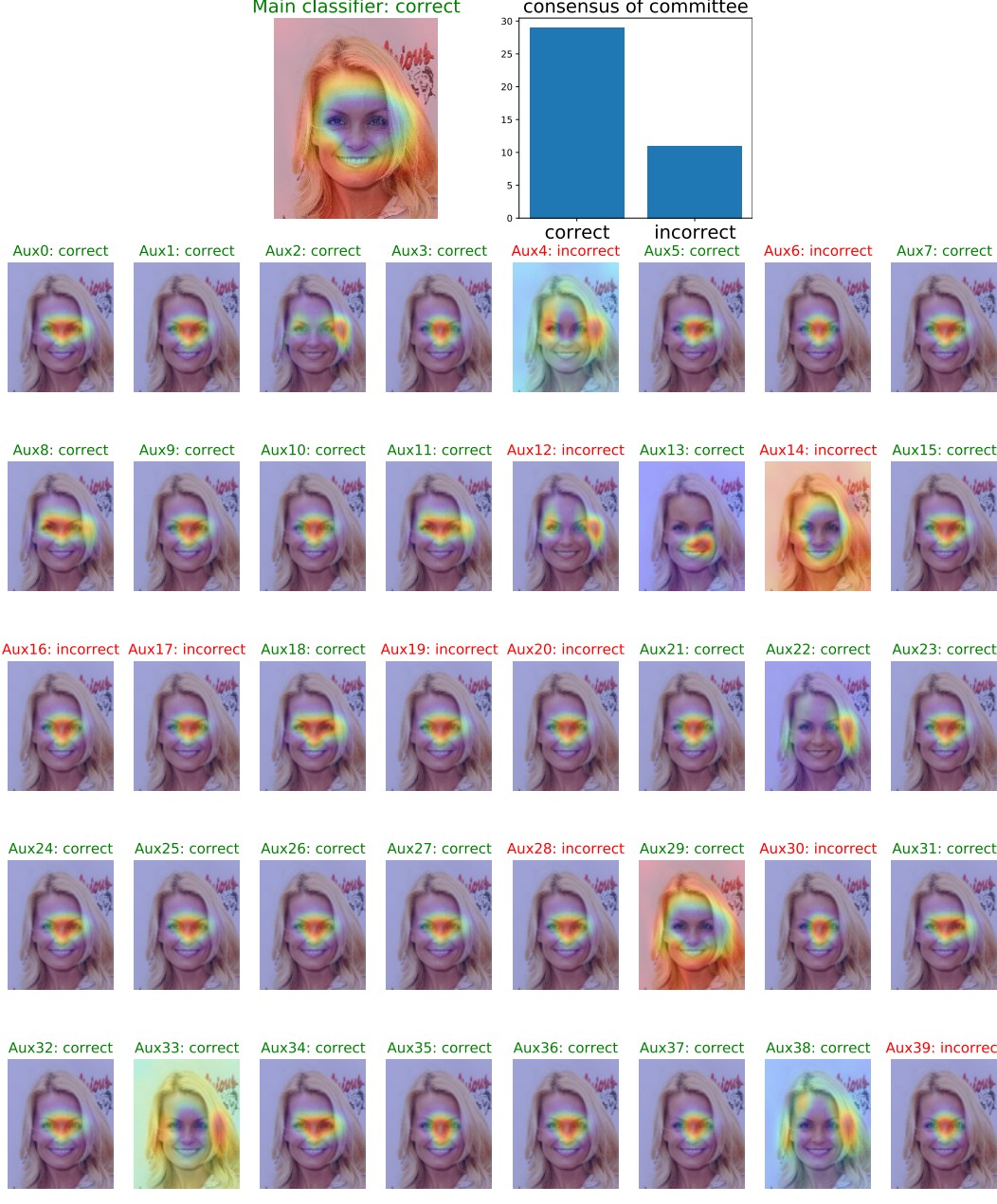

Figure 5: Class activation maps on a bias-guiding sample of CelebA and consensus graph. We mark a classifier that correctly predicts the class of the sample in 'correct', otherwise 'incorrect'.

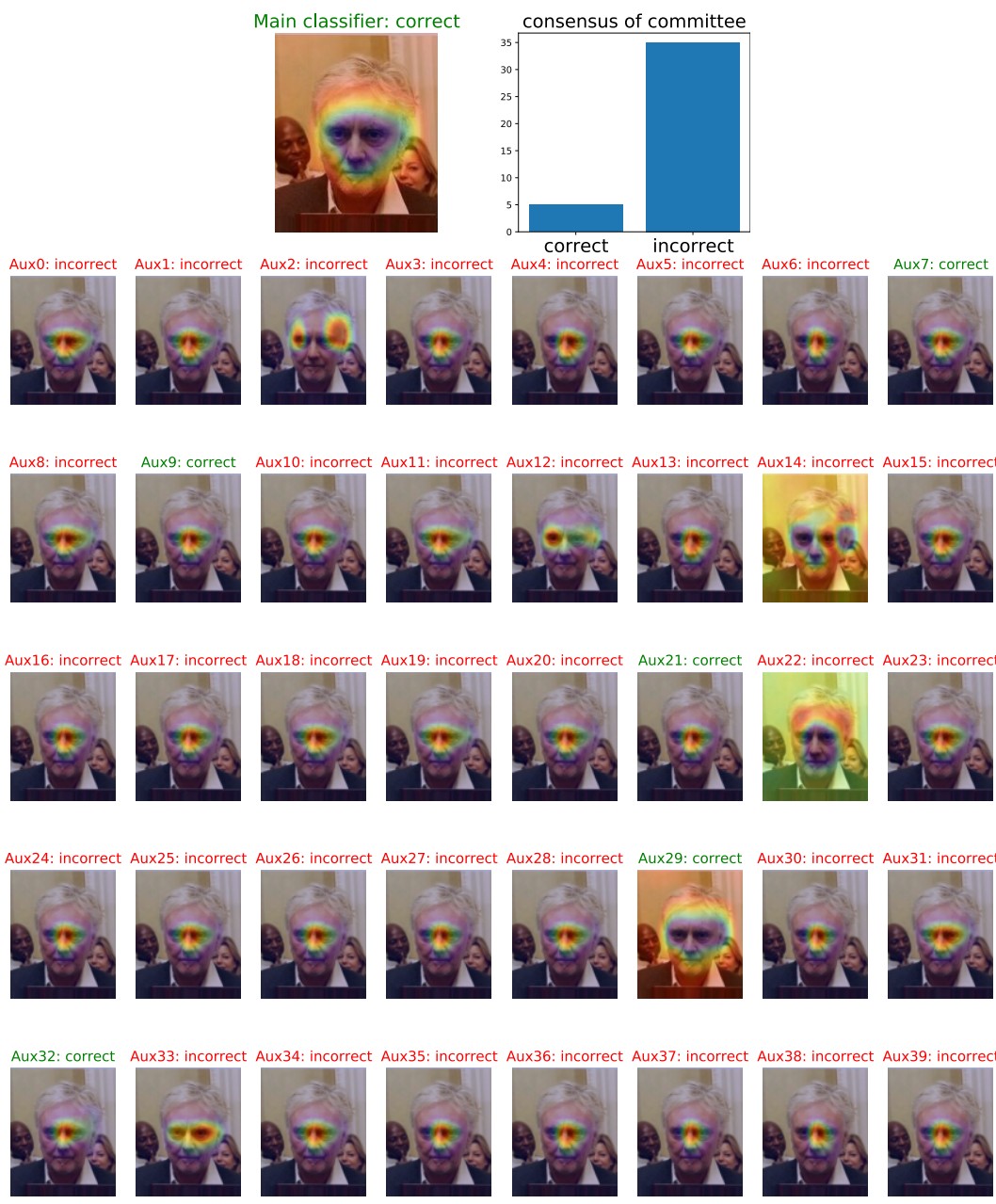

Figure 6: Class activation maps on a bias-conflicting sample of CelebA and consensus graph. We mark a classifier that correctly predicts the class of the sample in 'correct', otherwise 'incorrect'.

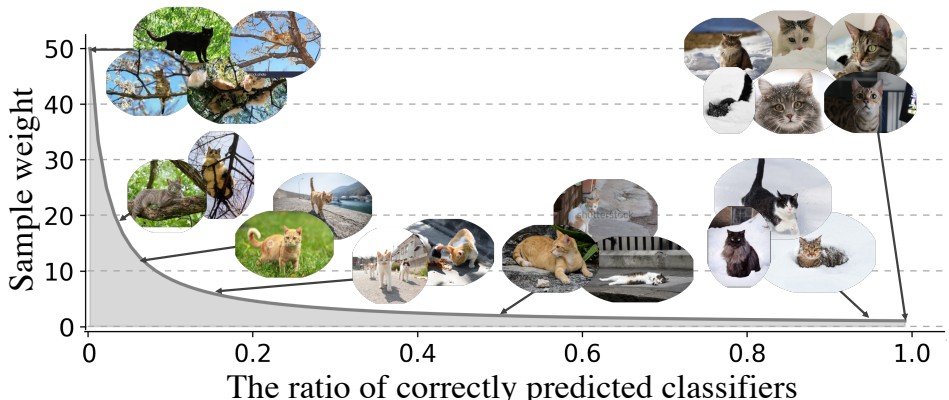

Figure 7: Qualitative examples on cat class of NICO. `Species` is target and `Context` is the bias. A majority of `cat` images on training set have `on snow` or `at home` context.

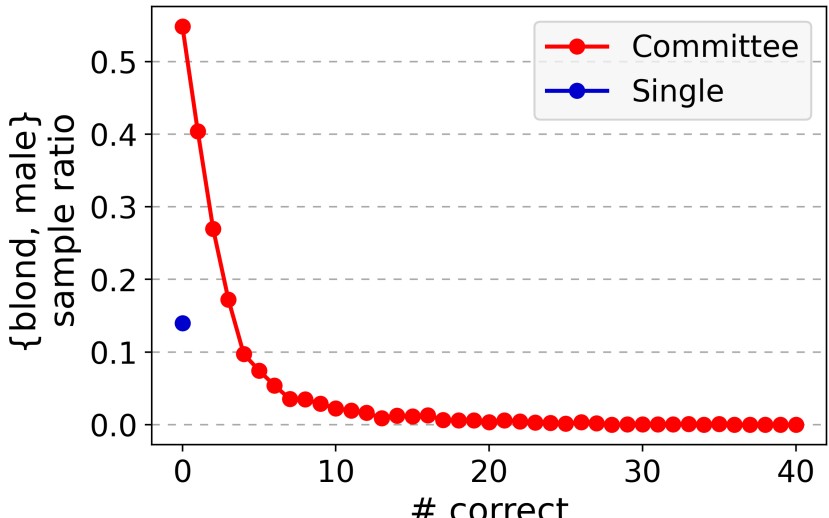

Figure 8: The ratio of bias-conflicting examples among the examples identified by a biased committee (red) and that of a single biased classifier (blue)