# OpenReview forum: "Learning Debiased Classifier with Biased Committee"
_NeurIPS.cc/2022/Conference — NeurIPS 2022 Accept_

### Official Review · Reviewer_Ff7t · 2022-07-05

**Rating:** 6
**Confidence:** 4
**Soundness:** 3 good
**Presentation:** 3 good
**Contribution:** 3 good

**Summary:**

The paper proposes a new approach to learning a debiased classifier with no spurious attribute label. Samples are weighted according to the consensus. The main classifier and committee classifier interacts. Committee classifier uses information about samples difficult to classify by the main classifier as distilled knowledge and integrates this into the loss function used to train the committee classifiers. Self-supervised representation obtained by BYOL is used as backbone.

**Questions:**

- BYOL is trained on the target dataset. It would be interesting to see how the representations learned this way on a limited data would compare to those learned from DINO (base or small VIT) pretrained on ImageNet.

- It looks like different hyperparameters are used for each dataset. How are these (learning rate, batch size, committee size, \lambda, \tau  etc.} optimized?

- LBC outperforms SSL+ERM on the CelebA dataset but is on par with it on the ImageNet (as well as BAR and NICO) datasets. Is this because bias is not as strong in texture-heavy images of ImageNet as in CelebA dataset?

- In the light of the limitations stated in the paper how can you ensure the improvement in CelebA dataset is real and is not due to fluctuations of randomness in the experimental process that affect smaller datasets more?

- Lines 206-207: “Therefore, we use HairColor as the target attribute and Gender as a spurious attribute, the same as HeavyMakeup.” This sentence is confusing. The proposed approach is not supposed to use the explicit knowledge of attributes causing bias. It is not clear  what it means to select these attributes as “target” and “spurious” attributes and how this information was used during training.


Corrections
Line 160: We identity

**Limitations:**

Algorithmic limitations are explained but societal impact is not.  What happens if the proposed algorithm fails to debias the data? This should have some ethical implications.

-------------------------------
Authors responses address my main concerns. As a result I update my score to Weak Accept.

**Strengths And Weaknesses:**

- Distilling the knowledge of difficult samples from the main classifier to train committee classifiers is an intriguing idea. Doing this with a self-supervised backbone is also carefully considered.

- Random subsets of training may not necessarily be dominated by bias-guiding samples especially in long-tailed datasets

- Figure 1 does not demonstrate that a committee of classifiers focus more on bias-conflicting samples. Higher enrichment can possibly be achieved due to other factors as well?

---

> ### Author Response · Authors · 2022-08-02
> **Response to Reviewer Ff7t (1/2)**
>
> We sincerely appreciate your insightful feedback and inspiring suggestions that helped improve our paper substantially. All the suggestions will be incorporated to the main text and the appendix. Please find our detailed responses to the comments below.
> ___
> **Q1. Random subsets of training may not necessarily be dominated by bias-guiding samples especially in long-tailed datasets.**
>
> A1. This is indeed an inspiring comment. We first note that in the community spurious correlation has been considered as a class-specific notion (i.e., correlation between a specific class and irrelevant attributes). Hence, bias-guiding samples will still dominate in a random subset "per class" and LBC will work as intended accordingly even in the case of long-tailed training data.
> However, class imbalance is another factor that ruins generalization of learned classifiers in many application scenarios, and we believe considering spurious correlation and class imbalance at the same time will pioneer a new direction towards learning robust and fair models in the wild.
> ___
> **Q2. Higher enrichment can possibly be achieved due to other factors as well?**
>
> A2. First of all, we apologize for the missing definition of Enrichment in the caption of Figure 1; it has been defined in Appendix A.2, which however should move to the main text for clarity. Enrichment, which is defined by
> $$
> \text{Enrichment}=\frac{(\text{sum of weights of bias-conflicting samples}/\text{sum of weights})}{(\text{\\# of bias-conflicting samples}/\text{\\# of samples})},
> $$
> consists of two factors: (1) the ratio of the sum of weights of bias-conflicting samples from the sum of weights, and (2) the ratio of the number of bias-conflicting samples from the number of samples. Because the ratio of bias-conflicting samples is a fixed value for each dataset, the ratio of weights of bias-conflicting samples is the only factor that determines the rank of an Enrichment value. Therefore, high Enrichment means that the model well emphasizes bias-conflicting samples among the identified samples by a model.
> ___
> **Q3. Comparison between LBC with SSL features pretrained on target dataset and that with SSL features pretrained on ImageNet.**
>
> A3. We demonstrate the impact of self-supervised learning on the target dataset by comparing our method with its variant using ImageNet-pretrained ResNet18 in Appendix A.3. We believe using DINO will further enhance performance of LBC thanks to its improved representation power.
> ___
> **Q4. Unclear protocol for choosing hyperparameters.**
>
> A4. We tune the hyperparameters using a validation set for every experiment, where we directly follow the existing validation set construction procedure. To be specific, the validation set is uniformly sampled from the whole dataset (ImageNet-9 and BAR), includes bias-guiding samples as much as bias-conflicting ones (NICO), or consists only of bias-conflicting ones (CelebA). In Table R1, the search spaces and performance metric used for the hyperparameter tuning are summarized. We will make these details publicly available along with our implementation of LBC.
>
> **[Table R1]**
> |Dataset|metric|set|learning rate|$m$|$\alpha$|$\|S_l\|$|$\lambda$|$\tau$|batch size|
> |---|---|---|---|---|---|---|---|---|---|
> |CelebA|CONFLICTING|validation|{0.01,0.004,0.005,0.006}|{10,20,30,40,50,60,70}|{0.015,0.02,0.025}|{200,300,400,600,800,900,1600}|{0.3,0.4,0.5,0.6,0.7,0.8}|{1,1.5,2,2.5,3}|256|
> |ImageNet-9|VALIDATION|validation|{0.001,0.0001}|{20,30,40}|{0.02,0.2}|{80}|{0.5,0.6,0.7,0.8,0.9}|{1}|128|
> |BAR|CONFLICTING|validation|{0.001,0.0001}|{20,30,40}|{0.02,0.2}|{10,20,30}|{0.5,0.6,0.7,0.8,0.9}|{1}|64|
> |NICO|VALIDATION|validation|{0.001,0.0001}|{20,30,40}|{0.02,0.2}|{10,20,30}|{0.5,0.6,0.7,0.8,0.9}|{1}|64|
> ||
> ___
> **Q5. Why is the LBC on par with SSL+ERM on some datasets**
>
> A5. Thanks for the detailed comment. The improvement of LBC over SSL+ERM is especially large on CelebA. One possible reason is that the SSL model is already robust enough to the bias of datasets other than CelebA. The augmentation strategy used in SSL, e.g., Gaussian blur, color jittering, and random cropping, may give features invariant to the bias attributes such as texture (ImageNet-9) and background (NICO, BAR). On the other hand, in the case of CelebA, using the SSL features directly for classification may be suboptimal since the model needs to focus on such attributes to correctly classify target labels (HairColor, HeavyMakeup). Thus, LBC could give a substantial performance boost by guiding the SSL features on CelebA. In this regard, we believe CelebA presents more realistic application scenarios than the other datasets since it is practically impossible to design the augmentation strategy for bias attributes that are unknown in training. Further analysis on when the SSL enhances robustness could be interesting future work.

---

> > ### Author Response · Authors · 2022-08-02
> > **Response to Reviewer Ff7t (2/2)**
> >
> > **Q6. The improvement on CelebA is not due to fluctuations of randomness**
> >
> > A6. We trained three models using our method through three independent trials with random seeds, and reported the degree of performance fluctuation in the form of standard deviation as well as the average accuracy of the three models. As shown in Table 1 and Table 2 of the main paper, although the performance fluctuation of our method on CelebA is not very small, it is comparable to that of prior work using no bias label like ours. We also would like to emphasize that the fluctuation is small enough on the ImageNet-9, BAR, and NICO datasets. Hence, we respectfully argue that the improvement of LBC on CelebA is not due to the performance fluctuation.
> > ___
> > **Q7. Clarification for Line 206-207: Selecting the target and spurious attributes for evaluation**
> >
> > A7. We do not use the information about the target and spuriously correlated attributes during training. We have access to bias labels that are manually annotated  only for evaluating how much a model is debiased from spurious correlation. Please note that all existing methods follow the same protocol. We will clarify the problem settings in the main text.
> > ___
> > **Q8. Ethical implications**
> >
> > A8. We apologize for missing the discussion on societal impact and will add the following discussion to the main text.
> > “Spurious correlation in training data leads to ethical issues (e.g., a model is degraded on a set of specific gender, ethnicity, or age of given input), and debiasing methods using no bias label like ours resolve the issues without human intervention. Due to the absence of bias labels in the problem setting, however, these methods do not explain whether a training dataset involves ethical issues and if the issues have been resolved by them.”
> > ___
> > **Typos**
> >
> > We appreciate your thoughtful comments. All the typos pointed out in the review have been fixed.

---

### Official Review · Reviewer_bx4Y · 2022-07-10

**Rating:** 6
**Confidence:** 5
**Soundness:** 3 good
**Presentation:** 3 good
**Contribution:** 3 good

**Summary:**

The paper proposes LBC to mitigate biases without bias labels. Motivated by the limitation of LfF’s single biased classifier, LBC overcomes this limitation by learning a committee of biased classifiers. The LBC is also trained with the knowledge of the main classifier through knowledge distillation. Furthermore, the paper explored the role of self-supervised representation in bias mitigation. The experiments are conducted on multiple datasets, showing better performance of the proposed method.

**Questions:**

Questions
* Clarify the inconsistent results of ERM and LfF between Table 1 and Table 2.
* Clarify the split used on CelebA.
* Clarify if the new split of the BAR dataset will be released for reproducibility.


Suggestions
* Cite and add discussion on other debiasing methods without spurious attribute labels.


**Limitations:**

Yes. The limitations are discussed in Sec. 5.

**Strengths And Weaknesses:**

Strengths

* The paper is well-written (except for a few typos) and easy to follow.
* The paper is well-motivated by the limitation of LfF’s single biased classifier (Fig. 1 and Sec. 1). The proposed method uses a committee with multiple biased classifiers to address this problem well, which has a novel contribution to this field.
* Using self-supervised representation for bias mitigation is also a novel exploration.


Weaknesses
* A few methods should also be cited and discussed in Section 2.2: PGI (Ahmed et al. 2021), ARL (Lahoti et al. 2020), George (Sohoni et al. 2020), and SD (Pezeshki et al. 2021), which are all debiasing methods without using spurious attribute labels.
Could the authors clarify why the results of ERM and LfF differ between Table 1 and Table 2? For example, ERM’s unbiased accuracy in Table 1 is 70.25 but becomes 95.6 in Table 2.

* Could the authors clarify which split is used to report the results on CelebA? The paper mentioned that the configuration follows [31] (L203), which uses the validation set to report the results. However, the experiment also uses CelebA’s validation set to tune the hyperparameter (L452-453).

* Could the authors promise to release the new split on BAR (L220-221)? Otherwise, it will become hard for future methods to benchmark.

Typos

* L51, 54, 68, and caption of Fig. 3: LCB -> LBC
* L160: identity -> identify
* L188: celebA -> CelebA


Although I have a few questions about the paper, the proposed method is generally novel and performs well on various datasets. My current rating is “Borderline accept.” I am willing to raise my rating if the authors can address my concerns in the response.

Reference

Ahmed, Faruk, Yoshua Bengio, Harm van Seijen, and Aaron Courville. 2021. “Systematic Generalisation with Group Invariant Predictions.” In International Conference on Learning Representations. https://openreview.net/forum?id=b9PoimzZFJ.

Sohoni, Nimit S., Jared A. Dunnmon, Geoffrey Angus, Albert Gu, and Christopher Ré. 2020. “No Subclass Left Behind: Fine-Grained Robustness in Coarse-Grained Classification Problems.” In Advances in Neural Information Processing Systems. http://arxiv.org/abs/2011.12945.

Lahoti, Preethi, Alex Beutel, Jilin Chen, Kang Lee, Flavien Prost, Nithum Thain, Xuezhi Wang, and Ed Chi. 2020. “Fairness without Demographics through Adversarially Reweighted Learning.” In Advances in Neural Information Processing Systems. Vol. 33. https://papers.nips.cc/paper/2020/hash/07fc15c9d169ee48573edd749d25945d-Abstract.html.

Pezeshki, Mohammad, Sékou-Oumar Kaba, Yoshua Bengio, Aaron Courville, Doina Precup, and Guillaume Lajoie. 2021. “Gradient Starvation: A Learning Proclivity in Neural Networks.” In Advances in Neural Information Processing Systems. https://openreview.net/forum?id=aExAsh1UHZo.

---

> ### Author Response · Authors · 2022-08-02
> **Response to Reviewer bx4Y**
>
> We sincerely appreciate your insightful feedback and valuable suggestions that helped improve our paper substantially. All the suggestions will be incorporated and the discussion on missing references will be added to the main text and the appendix. Please find our detailed responses to the comments below.
>
> **Q1. Clarification for the inconsistent results of ERM and LfF between Table 1 and Table 2**
>
> A1. The results of ERM and LfF are different between Table 1 and Table 2 due to the difference in the backbone network used in the experiments: All existing methods in Table 1 are tested with ResNet18 backbone, and those in Table 2 are tested with ResNet50 backbone. Please note that LBC incorporates the same backbone network, ResNet18, in both tables; we will make this clearer in the main text.
> ___
> **Q2. Clarification for the split used on CelebA**
>
> A2. We apologize for the confusion. We report test set performance on the table. All hyperparameters are tuned using a validation set.
>
> ___
> **Q3. Releasing the new split of the BAR dataset for reproducibility**
>
> A3. Thank you for your suggestion. We anonymously upload the new split of the BAR dataset on google drive. Please find the information from the BAR folder at https://url.kr/nqey9x. We promise releasing this publicly after the reviewing period.
> ___
> **S1. Missing related works about debiasing methods without spurious attribute labels**
>
> A1. Thanks for pointing out the missing references! We will cite them and add the following discussion on the papers in the main text.
>
> * Ahmed et al. 2021 proposed an invariance regularizer that matches the class-conditioned distribution of features across the groups.
> * Lahoti et al. 2020 proposed adversarially reweighted learning.
> * Sohoni et al. 2020 automatically identified unlabeled groups by clustering the feature vectors.
> * Pezeshki et al. 2021 decoupled learning dynamics to avoid capturing only a subset of task relevant features.
> ___
> **Typos**
>
> We appreciate your thoughtful comments. All the typos pointed out in the review have been fixed.

---

> > ### Comment · Reviewer_bx4Y · 2022-08-07
> > **Concerns Addressed**
> >
> > The authors addressed my concerns. I raised my rating to "6: Weak Accept."

---

### Official Review · Reviewer_Diuc · 2022-07-15

**Rating:** 6
**Confidence:** 4
**Soundness:** 3 good
**Presentation:** 4 excellent
**Contribution:** 3 good

**Summary:**

This paper considers the problem of learning to classify from spuriously-correlated data when annotations for the spurious attributes are absent. The authors propose to use the disagreement between an ensemble of classifiers trained on random subsets of the data (bagging)
to generate sample-weights for the classification loss. In addition to being trained on their respective subsets with supervision
from the ground-truth labels, they are provided with an additional error signal from the remaining data in the form of a distillation loss
with respect to the 'main' classifier, trained using the full dataset. The authors make use of self-supervision (BYOL) to train the backbone, on top of which the auxiliary classifiers are trained, finding this alone to enhance robustness and complementary to the aforementioned reweighting. The proposed method, LBC, is evaluated on five datasets exhibiting with known spurious correlations and compared against a variety of baselines that are both supervised and unsupervised w.r.t. the spurious attribute. Qualitative results are additionally illustrated for the CelebA dataset to provide insight into the sample weights and whether they align with the biases in the dataset.


**Questions:**

- Is the lack of a stop-gradient on the main classifier, $g$, intentional? Generally one doesn't allow the gradient to propagate through to the teacher network when performing knowledge distillation. If this is intentional, was it the result of testing or is there some theoretical justification for it?
- How essential is bagging? Could different random initialisations alone provide sufficient diversity, as suggested by the literature on deep ensembles?
- Why does the performance degrade past 40 ensemble members, rather than just plateauing?
- How important is the warm-up and how many iterations of it are typically required for stable performance?


**Limitations:**

I am generally satisfied with the extent to which the limitations of the proposed method have been discussed.

**Strengths And Weaknesses:**

### Strengths
- The idea of using a disagreement between ensemble members to generate sample-weights with the view to
improve fairness seems to be relatively novel
-  The authors do a good job of setting up the problem, establishing sufficient context for it in light of previous work, and motivating the use of a committee for identifying bias-conflicting samples through the analysis conducted in Figure 1.
- The proposed method is simple, intuitive, and demonstrably effective at alleviating spurious correlations.
- Evaluation was conducted using a good range of relevant datasets and baseline methods.
- The paper is well-presented, in that the results are easy to interpret, the proposed method can be understood based on its visual representation in Figure 2 alone, and the pseudocode is sufficiently detailed to make clear the more technical elements.
 - The paper is generally generally clear and well-written/structured, though there are some moderate grammatical errors, such as on line 225 (elaborated on below).
 - Good set of ablation studies demonstrating the contribution of each component of LBC, in turn, as well as the relationship
 between the size of the committee and performance. That said, it would be of interest to ablate the self-supervised component
 on its own to garner insight into how complementary it is to the proposed method, particularly as the dataset-task combination on which
 the ablations were performed show worse performance for SSL+ERM than ERM on its own (by a significant margin). Also of interest would be  how LBC compares with deep ensembling, where all ensemble members are trained on the full dataset with no distillation component.
 - Results are computed as the average over several replicates, alongside the with associated error intervals.

### Weaknesses
- The claim that this is the first paper to show the relationship between self-supervision and debiasing isn't entirely justified.
There are, for instance, [1] showing that large-scaling pre-training with SWAV improves fairness on downstream computer vision tasks,
and [2] showing that models trained with self-supervision are more robust to adversarial perturbations and label corruptions.
- Figure 4 isn't referenced in the main text. There should be some discussion as to what is shown here and whether it aligns with expectation (which it does seem to given that disagreement is highest for those samples from the minority group - Blond-Male).
- While the related work section is reasonably thorough, there is a gap when it comes to discussion
ensemble-based-debiasing methods (e.g. [3, 4]) and ensembling in the context of robustness generally, having obvious relevance to the proposed method.
- Despite the motivation encapsulated in Figure 1, there is a lack of theoretical guarantees regarding why and when should this bootstrapping approach should work
for spurious correlations in general.
- The term 'enrichment', as used in Figure 1, is not explicitly defined.

### Misc.
- Liu et al. (Just Train Twice) is cited using two different references.
- 'Self sup' should be SSL in the table of ablations for consistency with 'SSL+ERM'.
- The parenthetical remark "e.g., most dog images are appeared ‘on grass’ the others are appeared on 6 context" doesn't make complete sense -- I assume this is supposed to read something to the effect of "More dog images appear on the 'on grass' context than any of the other 6 contexts"

---

> ### Author Response · Authors · 2022-08-02
> **Response to Reviewer Diuc (1/2)**
>
> We sincerely appreciate your insightful feedback and constructive suggestions that helped improve our paper substantially. All the suggestions will be incorporated, and additional discussions will be added to the main text. Please find our detailed responses to the comments below.
>
> **Q1. Lack of justification for claiming to be the first paper to show the relationship between self-supervision and debiasing**
>
> A1. Thank you for the valuable comment. If you could kindly let us know the information about the papers [1,2], which is unfortunately missing in the current review, we will cite and discuss them in detail to more appropriately claim the novelty of our method.  Also, we will tone-down our claim by rephrasing it as "our method is the first attempt to employ self-supervised representation for extracting features less sensitive to spurious correlation."
> ___
> **Q2.Lack of discussions for Figure 4**
>
> A2. We will add the following explanation, which is originally given in Appendix A.4.2 along with additional figures, to the main paper.
> “Figure 4 shows the curve of the weight function in Eq. (2) and CelebA images with specific weight values; in CelebA, most ‘blond hair’ images on the training set are ‘woman,’ and ‘blond-haired male’ images are bias-conflicting samples.  As illustrated in the figure, our method not only up-weights the bias-conflicting samples and down-weights the bias-guiding samples but also assigns a weight to a sample precisely according to its difficulty”
> ___
> **Q3. Missing related works about ensemble-based-debiasing methods**
>
> A3. Thank you very much for pointing out the missing references. However, unfortunately, the papers [3,4] are not specified in the review; if you could kindly let us know the information about the papers, we will cite and discuss them in the main text during the discussion period.
> ___
> **Q4. Theoretical guarantees of bootstrapping approach for spurious correlations**
>
> A4. Thank you for the constructive comment. While our work focuses on empirical aspects of learning a debiased classifier, we agree that theoretical analysis may further provide insights. To this end, we first analyze the effects of the bootstrapping more rigorously.
>
> The bootstrapping allows the sampled subsets to have the following three properties.
> - Independence: The sampling with replacement strategy of the bootstrapping ensures that each subset is independent from the others. In other words, a subset is not affected by sampling history of the other subsets.
> - Diversity: Since the size of each subset is quite smaller than that of the entire dataset in our setting, the subsets are highly likely to be diverse. In other words, two of such subsets are not likely to be the same since the probability of sampling a specific subset, $n^{-s}$, where $n$ is the size of the entire training set and $s$ is the size of subsets, is very small as $n$ is usually quite large.
> - Being dominated by bias-guiding data: The expected number of bias-guiding data within such a subset is $rs$, where $r$ is the ratio of bias-guiding data and the entire data and $s$ is the size of subsets, since the process of sampling a bias-guiding example can be formulated as a Bernoulli trial with the success probability (the probability of sampling a bias-guiding one) $r$. Hence, it is expected that each bootstrapped subset is dominated by bias-guiding data as much as the entire training set is.
>
> Since the subsets are independent, diverse, and dominated by bias-guiding samples, classifiers of the committee trained using such subsets are also expected to be independent of each other, diverse in prediction, and biased towards spurious correlation. Please note that the independence and diversity of the classifiers are necessary conditions for the committee to be a wise crowd [Surowiecki 2004], and have been proven to be main factors behind the success of ensemble learning [Dietterich 2000]. Finally, it is obvious that each classifier should be biased for the purpose of the committee.
>
> [Surowiecki 2004] James Surowiecki. The Wisdom of Crowds. 2004.
>
> [Dietterich 2000] Thomas G. Dietterich. Ensemble Methods in Machine Learning. International workshop on Multiple Classifier Systems, 2000.
> ___
> **Q5. Lack of explicit definition for 'Enrichment'**
>
> A5. We apologize for the clarity issue. The definition and explanation of Enrichment are given in Appendix A.2, and will move to the main text in the revision. Enrichment is defined by
> $$
> \text{Enrichment}=\frac{(\text{sum of weights of bias-conflicting samples}/\text{sum of weights})}{(\text{\\# of bias-conflicting samples}/\text{\\# of samples})}.
> $$
> Because the ratio of bias-conflicting samples is a fixed value for each dataset, the ratio of weights of bias-conflicting samples is the only factor that determines the rank of an Enrichment value. Therefore high Enrichment means that the model well emphasizes bias-conflicting samples among the identified samples by a model.

---

> > ### Author Response · Authors · 2022-08-02
> > **Response to Reviewer Diuc (2/2)**
> >
> > **Q6. Clarification for stop-gradient on the main classifier $g$**
> >
> > A6. As with knowledge distillation methods, we also do a stop-gradient on the main classifier; this is described in line 12 of Algorithm 1 in the main text. We apologize for the confusion, and will add the explanation in the paper.
> > ___
> > **Q7. Performance degradation past 40 ensemble members**
> >
> > A7. Thank you for the detailed comment. We consider the performance after 40 ensemble members plateauing since the most significant performance drop past m=40 is 1.3%p, within the standard deviation of our final performance. The slight performance degradation may come from less optimal hyper-parameters compared to that of our final model. Please also note that, even after m=40, LBC still achieves the state of the art.
> > ___
> > **Q8. Details on warm-up iterations**
> >
> > A8. We warm-up the committee during 3 epochs on small datasets (NICO, BAR) and 5 epochs on large datasets (CelebA, ImageNet). We did not carefully tune the number of warm-up iterations and empirically found that 3~5 epochs were enough to achieve stable performance.
> > ___
> > **Miscellaneous**
> >
> > We have fixed them. We appreciate your thoughtful comments.

---

### Official Review · Reviewer_zA6x · 2022-07-18

**Rating:** 6
**Confidence:** 4
**Soundness:** 2 fair
**Presentation:** 3 good
**Contribution:** 3 good

**Summary:**

The paper proposes a method to de-bias classifiers and make them robust to spurious correlations, which is mainly based on learning an ensemble of intentionally biased models which will help to identify minority (bias-conflicting) examples in train data and increase their importance / weight while training a final de-biased model. Additional details of the method include (1) pre-training representations with self-supervised learning and learning shallow models on top of these representations, and (2) alternate updating of the biased ensemble and de-biased classifier where the knowledge is transferred from the classifier to the ensemble. The approach shows strong performance on a number of benchmarks: CelebA, ImageNet-9, ImageNet-A, BAR and NICO.

**Questions:**

Please see section "Quality" in "Strengths And Weaknesses".

**Limitations:**

Yes

**Strengths And Weaknesses:**

### Originality
The proposed method is novel, to the best of my knowledge. Although prior works proposed to use an internationally biased model to detect minority examples in the past (e.g., [1], [2]) this papers is the first to proposed a biased bootstrapped ensemble and shows that it increases robustness of the final classifier model. The paper is based on a combination of known techniques (bootstrapped ensemble, using auxiliary module to identify biases, self-supervised learning, knowledge distillation), however, the proposed approach shows strong performance in practice, and, thus, I believe is a valuable contribution.


### Quality

1. My main concern for the paper evaluation is regarding the hyperparameter tuning: the authors described which hyperparameters they picked in lines 192-195, but it is not mentioned how exactly they were tuned and which range of values was considered for each hyper-parameter. It was shown in prior works on spurious correlation robustness that the worst-group results may vary a lot depending on whether "in-distribution" or "out-of-distribution" data was used for hyper-parameter tuning (i.e. which portion of the validation set was bias-guiding vs bias-conflicting) and whether the hyper-parameters were chosen based on average accuracy or worst-group accuracy on validation (the latter would imply that we assume having access to bias labels for validation data). Also e.g. on Figure 3(b) it is shown that worst-group performance varies quite a bit depending on the number of classifiers in the committee and it is mentioned that $m$ was chosen to be the one corresponding to the best performance -- but it is not clear whether this number was chosen based on best performance on validation or test.

It would be helpful to know which assumptions on validation data the authors made and the exact strategy for hyper-parameter tuning.

2. While the observation that self-supervised (SSL) training improves the robustness to biases on most benchmarks is interesting, it sounds a bit disconnected from the rest of the paper. I think it is also not fair to compare SSL+LBC to other prior methods which were not using SSL pretraining (especially considering that SSL+ERM is already beating most methods on all benchmarks except CelebA). I think to perform a careful and fair ablation it would be needed to (1) compare LBC w/o SSL pretraining to prior works (to isolate the effect of improved representations), (2) with fixed SSL features, train and compare different prior methods and LBC on top of these features.

3. I found it a bit surprising that such small subsets of data $S_l$ were used to train individual networks in the committee, I guess it is related to the fact that the authors train quite shallow MLP models on top of SSL features. I would be really interested to see how important boostrapping is in general for the method, e.g. what if one swapped bootstrapped ensemble with regular deep ensemble? Would the performance change much in cases when SSL pretraining is used and when there is no SSL pretraining (so we are training not a shallow but deep model)? It would also be interesting to see how the two parameters the size of $S_l$ and the number of members in the committee interact: is it the case that we might get the same performance with the smaller number of networks in the committee if they are trained on bigger splits of train data?

4. It would be helpful for understanding of the method to see the plot similar to Figure 1(a) but for the proposed approach showing what the ratio of bias-conflicting examples among the examples identified by a biased committee is.

5. Since the method consists of a few independent parts (biased committee, SSL, knowledge distillation) it would be helpful to see more ablation studies varying just one of those components, e.g. how much would the KD be helpful when using only one biased model as opposed to the committee? Similarly, what would the performance be with a single biased model on SSL features? (That would be quite related to reproducing a method like JTT [2] on SSL featured as mentioned in point 3 above)

6. It would be interesting how this method works on other data modalities, e.g. NLP spurious correlation tasks such as CivilComments or MultiNLI.


### Clarity
The paper is clearly written and easy to follow. A minor note is that it might be more natural to swap 4.1 and 4.2 and first describe the datasets in the Experiments section since they are mentioned in 4.1.


### Significance
I think although the method is based on a combination of known techniques, it significantly advances the performance on robustness benchmarks and especially if the authors address the evaluation concern described above the paper will be interesting for the robustness community.


References

[1] Nam et al, Learning from Failure: Training Debiased Classifier from Biased Classifier

[2] Liu et al, Just Train Twice: Improving Group Robustness without Training Group Information

[3] Lee et al, Diversify and Disambiguate: Learning From Underspecified Data

---

> ### Author Response · Authors · 2022-08-02
> **Response to Reviewer zA6x (1/3)**
>
> We sincerely appreciate your insightful feedback and valuable suggestions that helped improve our paper substantially. All the suggestions will be incorporated and additional experiments will be added to the main text and the appendix. Please find our detailed responses to the comments below.
> ___
> **Q1. Unclear protocol for choosing hyperparameters**
>
> A1. For every experiment, we tune the hyperparameters using a validation set. For a fair comparison, we follow the existing validation set construction procedure, which varies for different datasets. To be specific, the validation sets are either uniformly sampled from the whole dataset (ImageNet-9 and BAR), include bias-guiding samples as much as bias-conflicting ones (NICO), or consist only of bias-conflicting ones (CelebA). The hyperparameters, their search spaces, and performance metric used for the tuning are summarized in Table R1. We will describe these details in the appendix and make them publicly available along with our implementation of LBC.
>
> **[Table R1]**
> |Dataset|metric|set|learning rate|$m$|$\alpha$|$\|S_l\|$|$\lambda$|$\tau$|batch size|
> |---|---|---|---|---|---|---|---|---|---|
> |CelebA|CONFLICTING|validation|{0.01,0.004,0.005,0.006}|{10,20,30,40,50,60,70}|{0.015,0.02,0.025}|{200,300,400,600,800,900,1600}|{0.3,0.4,0.5,0.6,0.7,0.8}|{1,1.5,2,2.5,3}|256|
> |ImageNet-9|VALIDATION|validation|{0.001,0.0001}|{20,30,40}|{0.02,0.2}|{80}|{0.5,0.6,0.7,0.8,0.9}|{1}|128|
> |BAR|CONFLICTING|validation|{0.001,0.0001}|{20,30,40}|{0.02,0.2}|{10,20,30}|{0.5,0.6,0.7,0.8,0.9}|{1}|64|
> |NICO|VALIDATION|validation|{0.001,0.0001}|{20,30,40}|{0.02,0.2}|{10,20,30}|{0.5,0.6,0.7,0.8,0.9}|{1}|64|
>
> ___
> **Q2-1. Comparison between LBC without SSL features and existing methods**
>
> A2-1. Thank you for suggesting important experiments that we have missed! We compare LBC without SSL features with existing methods as requested. As shown in Table R2 and R3, LBC without SSL features (LBC w/o SSL) outperformed existing methods with the same backbone network (ResNet18) and was as competitive as those with a substantially stronger backbone (ResNet50).
>
> **[Table R2]**
> |Method|Backbone network|CelebA GUIDING|CelebA UNBIASED|CelebA CONFLICTING|
> |---|---|---|---|---|
> |ERM|ResNet18|87.98|70.25$\pm$ 0.35|52.52$\pm$ 0.19|
> |LfF|ResNet18|87.24|84.24$\pm$ 0.37|81.24$\pm$ 1.38|
> |JTT|ResNet18|94.70|87.85|81.00|
> |LBC w/o SSL|ResNet18|90.29$\pm$ 1.64|87.11$\pm$ 0.57|83.93$\pm$ 1.55|
> |LBC w/ SSL|ResNet18|90.57$\pm$ 2.15|88.90$\pm$ 1.55|87.22$\pm$ 1.14|
>
> **[Table R3]**
> |Method|Backbone network|CelebA WORST-GROUP|
> |---|---|---|
> |ERM|ResNet50|47.2|
> |LfF|ResNet50|70.6|
> |EIIL|ResNet50|83.3|
> |JTT|ResNet50|81.1|
> |JTT|ResNet18|76.7|
> |LBC w/o SSL|ResNet18|80.8$\pm$ 1.6|
> |LBC w/ SSL|ResNet18|85.5$\pm$ 1.4|
>
> **Q2-2. Comparison between LBC and existing methods using SSL features**
>
> A2-2. We also compare our method with LfF and JTT using SSL features as requested in Table R4. Although LfF and JTT were also trained on SSL features, LBC outperforms them on all datasets.
>
> **[Table R4]**
> |method|CelebA WORST-GROUP|ImageNet-A TEST|BAR TEST|NICO TEST|
> |---|---|---|---|---|
> |ERM w/ SSL|38.5$\pm$ 4.1|34.21$\pm$ 0.49|35.32$\pm$ 0.46|52.24$\pm$ 0.27|
> |LfF w/ SSL|76.6$\pm$ 5.8|35.11$\pm$ 0.93|58.75$\pm$ 0.41|51.99$\pm$ 0.16|
> |JTT w/ SSL|57.5$\pm$ 11.5|33.39$\pm$ 0.16|59.37$\pm$ 2.57|52.04$\pm$ 0.27|
> |LBC w/ SSL|85.5$\pm$ 1.4|35.97$\pm$ 0.49|62.03$\pm$ 0.74|52.84$\pm$ 0.31|

---

> > ### Author Response · Authors · 2022-08-02
> > **Response to Reviewer zA6x (2/3)**
> >
> > **Q3-1. How important bootstrapping is in general for LBC?**
> >
> > A3-1. To demonstrate the contribution of bootstrapping in our method more clearly, we compare our method with deep ensemble, ensemble using SSL features, and ours without SSL features, as requested. The results are summarized in Table R5. Note that the deep ensemble consists of only 5 classifiers as it demands a substantially larger memory footprint than the others. In addition, the ensemble using SSL features is implemented as multiple classification heads appended on top of the SSL backbone. Deep ensemble showed the worst performance. Using SSL features and increasing the size of the ensemble improves performance slightly, but still largely inferior to LBC. As shown in Table R5, bootstrapping (i.e., LBC) substantially outperforms the naive ensemble regardless of the use of SSL features. In addition, using SSL features further improves performance.
> >
> > **[Table R5]**
> > ||Committee member|Training data for each member|SSL feature|m|WORST-GROUP|
> > |---|---|---|---|---|---|
> > |Deep ensemble|whole model|entire training dataset|&#x2717;|5|77.22|
> > |Ensemble w/ SSL|two FC layers|entire training dataset|$\checkmark$|40|78.0$\pm$ 4.3|
> > |LBC w/o SSL|two FC layers|random subset (bootstrapped)|&#x2717;|40|80.8$\pm$ 1.6|
> > |LBC w/ SSL|two FC layers|random subset (bootstrapped)|$\checkmark$|40|85.5$\pm$ 1.4|
> > ||
> >
> > **Q3-2. Performance of a smaller committee trained on bigger splits of train data**
> >
> > A3-2. Thank you for the important experiment idea! In Table R6, we report the WORST-GROUP accuracy on CelebA test set according to the number of classifiers and their subset size. The accuracy of LBC reported in the main paper is enclosed in quotation marks, and the best accuracy in each column or row is marked in bold. The accuracy is generally high enough when the set size is between 300 and 400 and the number of classifiers is over 40; the performance is not very sensitive to hyperparameters within the reasonable range of hyperparameter values. We also would like to note that, the accuracy reported in the main paper (enclosed in quotation marks) is lower than the best accuracy of this table since the hyperparameters are not tuned on the test set, but still achieves the state of the art on CelebA.
> >
> > **[Table R6]**
> > | # of classifiers \ set size | 10    | 50    | 100   | 200   | 300   | 400   | 500   | 600   |   |
> > |-----------------|-------|-------|-------|-------|-------|-------|-------|-------|---|
> > | 70              | 83.33 | 84.44 | 83.89 | 81.67 | 85.00 | **87.22** | 76.67 | 78.89 |   |
> > | 60              | **84.80** | 81.67 | 80.00 | 76.11 | 85.40 | **86.96** | 86.11 | 80.56 |   |
> > | 50              | 82.73 | **86.15** | **85.67** | **86.11** | 84.90 | **86.67** | **86.67** | 82.48 |   |
> > | 40              | 81.67 | 82.22 | 83.33 | 84.44 | **"85.5"** | 82.67 | **87.80** | 83.89 |   |
> > | 30              | 70.93 | 82.78 | 76.11 | 86.27 | 81.10 | **87.22** | 85.28 | 81.37 |   |
> > | 20              | 82.78 | 83.33 | 85.00 | 81.67 | 84.50 | 82.10 | 83.08 | **85.04** |   |
> > | 10              | 76.11 | 72.78 | 80.56 | 78.19 | 82.80 | 82.44 | **84.18** | 39.48 |   |
> > ||
> >
> > ___
> > **Q4. Plot for the ratio of bias-conflicting examples among the examples identified by a biased committee**
> >
> > A4. We added the requested plot in Appendix A.5, and for your convenience, we present the content in the form of Table R7. Table R7 shows that the ratio of bias-conflicting samples to the whole identified samples is much higher when using the committee, which suggests that the committee identifies bias-conflicting samples more precisely than a single biased classifier.
> >
> > **[Table R7]**
> > |# of correct biased classifiers|0|1|2|3|4|5|6|7|8~9|10~12|13~21|22~40|
> > |---|---|---|---|---|---|---|---|---|---|---|---|---|
> > |biased committee|0.55|0.4|0.27|0.17|0.1|0.07|0.05|0.04|0.03|0.02|0.01|0.00|
> > |single biased classifier|0.14|-|-|-|-|-|-|-|-|-|-|-|
> > ||

---

> > > ### Author Response · Authors · 2022-08-02
> > > **Response to Reviewer zA6x (3/3)**
> > >
> > > **Q5. Additional ablation study**
> > >
> > > We greatly appreciate your valuable suggestions and extend Table 5 of the main paper to address the suggestions. The extended table can be found below (Table R8).
> > >
> > > **How much would KD be useful when using a single biased model**: To incorporate your comment, we first conduct two additional ablation studies with two variants of LBC to figure out the effectiveness of KD. The first variant incorporates a single biased classifier learned on a subset of the training set (row 4), and the second variant additionally adopts KD (row 5). The results in Table R8 suggest that KD is useful for debiasing regardless of the use of the committee. Table R8 now even more clearly shows that every component of LBC contributes to debiasing.
> > >
> > > **Performance with a single biased model on SSL features**: As demonstrated in Table R8, every variant of LBC using SSL features outperforms JTT using SSL features on CelebA. The performance of the LBC variant using a single biased model trained on the whole training set (row 3) significantly outperforms that of JTT on SSL features (row 2). The main difference between the two models, which leads to the performance gap, is two fold. First, they use different weight functions. Second, the single biased classifier of the LBC variant produces sample weights at every iteration while the weights are computed once and fixed during training in the JTT variant.
> > >
> > > **[Table R8]**
> > > |SSL|Weight function|Training data for each committee member|Committee size (m)|KD|CelebA WORST-GROUP|
> > > |---|---|---|------|---|---|
> > > |$\checkmark$|ERM|-|0|&#x2717;|38.5 $\pm$ 4.1|
> > > |$\checkmark$|JTT|entire training dataset|1|&#x2717;|57.7 $\pm$ 11.5|
> > > |$\checkmark$|Ours|entire training dataset|1|&#x2717;|64.1 $\pm$ 2.4|
> > > |$\checkmark$|Ours|random subset (bootstrapped)|1|&#x2717;|72.7 $\pm$ 11.7|
> > > |$\checkmark$|Ours|random subset (bootstrapped)|1|$\checkmark$|77.9 $\pm$ 3.1|
> > > |$\checkmark$|Ours|entire training dataset|40|&#x2717;|78.0 $\pm$ 4.3|
> > > |$\checkmark$|Ours|random subset (bootstrapped)|40|&#x2717;|81.3 $\pm$ 2.3|
> > > |$\checkmark$|Ours|random subset (bootstrapped)|40|$\checkmark$|85.5 $\pm$ 1.4|
> > > ||
> > >
> > > \* The search space of hyperparameters of JTT on SSL features is as follows: {1,2, 10, 20, 30, 50} for the identification epoch and {20, 50, 100} for the upweight value. Optimal hyperparameters we found are epoch=10 and upweight value=20.
> > >
> > > ---
> > > **Q6. Extension to other data modalities (e.g., NLP spurious correlation tasks)**
> > >
> > > A6. Thanks for the valuable suggestion! LBC can be applied to data of any modalities as long as they can be represented as feature vectors, and applying LBC to spurious correlation tasks in the NLP domain is definitely next on our agenda.

---

> > > > ### Comment · Reviewer_zA6x · 2022-08-08
> > > > **Response**
> > > >
> > > > Thank you for the detailed answers to my questions and additional experiments! I hope the authors will include the ablation experiments results into the next revision of the paper.
> > > >
> > > > A few minor comments:
> > > >
> > > > Table R3. For clean comparison I would encourage the authors to fix the architecture while varying the method. In the next revision, please change the architecture to ResNet50 for LBC in Table R3 of the rebuttal and Table 2 of the main paper.
> > > >
> > > > Table R5: I would encourage the authors to include ablation with the vanilla deep ensembles with more networks (m>5) in the next version of the paper.
> > > >
> > > > Table R6: Thank you for adding this experiment. The set size 600 is still a very small subset of CelebA, and it would be interesting to see the performance with larger $S_l$ in the future version of the paper.
> > > >
> > > > Table R7: It is not clear what the columns correspond to in the Table (or what x-axis means in Figure 10 of Appendix A.5), please clarify that in the updated version.
> > > >
> > > > I’ve read the other reviews and responses to the reviews, and I think the paper is overall providing a useful contribution. I maintain my original “weak accept” score.

---

### Meta-Review · Area_Chair_tPJy · 2022-08-24

**Recommendation:** Accept
**Confidence:** Certain

**Metareview:**

The manuscript considers the problem of learning to classify from spuriously-correlated data when annotations for the spurious attributes are absent. The general idea is to learn an ensemble of intentionally biased models which will help to identify minority (bias-conflicting) examples in train data and increase their importance / weight while training a final de-biased model. Additional details of the method include (1) pre-training representations with self-supervised learning and learning shallow models on top of these representations, and (2) alternate updating of the biased ensemble and de-biased classifier where the knowledge is transferred from the classifier to the ensemble. The approach shows strong performance on five datasets: CelebA, ImageNet-9, ImageNet-A, BAR and NICO. Qualitative results are additionally illustrated for the CelebA dataset to provide insight into the sample weights and whether they align with the biases in the dataset.

Reviewers acknowledged several positive aspects of the manuscript including the proposed idea of distilling the knowledge of difficult samples from the main classifier to train committee classifiers. Performing this with a self-supervised backbone was also carefully considered. The manuscript is based on a combination of known techniques (bootstrapped ensemble, using auxiliary module to identify biases, self-supervised learning, knowledge distillation), however, the proposed approach shows strong performance in practice.

Discussion phase has addressed many concerns related to experimental evaluations inc. hyperparameter tuning, importance of self-supervised backbone, and importance of bootstrapping.


**Award:**

No

---

### Decision · Program_Chairs · 2022-09-14

Accept